# ICE-CAMERA: a flatbed scanner to study inland Antarctic polar precipitation.

Massimo Del Guasta

Istituto Nazionale Ottica CNR, Sesto Fiorentino, 50019, Firenze, Italy

*Correspondence to*: Massimo Del Guasta (massimo.delguasta@ino.cnr.it)

**Abstract.**

Studying precipitation at very high latitudes is difficult because of the harsh environmental conditions that limit the external activity of humans and instruments, especially in the polar winter. The direct monitoring of ice crystal habits and size distribution in antarctic precipitation is important for the validation of the algorithms used for retrieving precipitation from ground-based and satellite-borne radar instruments, and for the improvement of the climatological modeling of polar areas. The paper describes an automated device (ICE-CAMERA) specifically developed for the imaging, measurement and classification of ice precipitation on the Antarctic high plateau. The instrument gives detailed information on precipitation on an hourly basis. The article provides a description of the device and its image processing software. Starting in 2014, the instrument operates *almost* unattended all year round at Concordia station, Antarctica (75°S, 123°E, 3220 m altitude).

## 1. Introduction.

In Antarctica, the characteristics of ice precipitation depend greatly on the region. In coastal areas, precipitation is influenced by synoptic scale features, such as cyclones and fronts (Bromwich, 1988). In the interior (> 2500 m), a significant part of the precipitation falls in the form of small ice crystals ("diamond dust", DD) under clear-sky conditions (Fujita and Abe, 2006). Snow particles over Antarctica are generally smaller compared to other regions of the world. The largest particles are found close to the coast, where more water vapour is available and diameters up to 10 mm are recorded (Konishi et al., 1992) with particle shapes similar to mid-latitude ones (Satow, 1983). Most of the bigger particles are aggregates (some can be found in the dataset of Grazioli et al.,2022). More inland stations record snowflakes of much smaller sizes, ranging from particles smaller than 100 μm at South Pole (Walden et al., 2003, Lawson et al., 2006) till hundreds of μm at other inland stations (Lachlan-Cope et al., 2001).

In situ measurements of precipitation are rare in Antarctica and affected by large uncertainties. This is particularly true in the high plateau, where less than 20 cm of snow accumulates every year (Palerme et al.,2014). As a result, the global precipitation products that rely on these observations (i.e. the Global Precipitation Climatology Centre (GPCC), (Schneider et al., 2017)) have no coverage over this region. Other observational products, such as the Global Precipitation Climatology Project (GPCP) (Huffman et al.,2001) , that uses GPCC for bias correction over land, has relied on satellite-only precipitation estimates. Satellite products also face large uncertainties over cold regions such as Antarctica due to insufficient sensitivity of sensors to detect and estimate precipitation signals, complex surface emissivities, and poor understanding of precipitation microphysics. Ground based K-band radars (~1-cm wavelength) are robust instruments successfully employed for studying precipitation in coastal Antarctic sites (Souverijns et al., 2017) but are quite blind to the sub-millimetre ice particles encountered on the plateau, due to the relationship $D^6$ between the radar scattering cross-section and the particle diameter (D).

 The satellite-borne radar CloudSat (Liu, 2008) did provide a quantum leap in observing ice in the Antarctic atmosphere (up to 82 °S), but being a single-frequency radar (like K-band radars), the retrieval of precipitation quantities relies on many assumptions about the properties of particles, resulting into ±50% uncertainties for IWC (Heymsfield et al.,2008). The microphysical assumptions (shapes and size distribution of particles) are the biggest causes for IWC, IWP, and snowfall rate retrieval uncertainty (Hiley et al.,2011, Wood et al.,2015). Moreover, CloudSat bins close to the ground cannot be used for precipitation retrieval, resulting into a severe underestimation of the diamond-dust and blowing-snow contribution to Antarctic snow balance (Palm et al., 2018). Despite these uncertainties, in absence of ground validation CloudSat data are now used as

independent dataset for the validation of precipitation models in Antarctica (Palerme et al.,2014, Palerme et al.,2017).
The direct observation and the continuous monitoring of habit and size distribution of precipitation is therefore required in
order to validate both precipitation models, CloudSat and radar algorithms on the Antarctic plateau.
Disdrometers are robust *in-situ* devices, increasingly used in Antarctic coastal areas (Souverijns et al., 2017, Bracci et al.,2022).
They provide the size distribution and falling speed of hydrometers, but they give no direct information about the shape. The
evolution of disdrometers into 2D-disdrometers gave access to some shape indications about hydrometeors (Grazioli et
al.,2014). A further evolution of disdrometers into Imaging-disdrometers, such as the Snowflake Video Imager (SVI) (Newman
et al., 2009), provided realistic images of the crystals. Grazioli et al., (2017), as part of a multidisciplinary field campaigns,
deployed a multi-angle snowflake camera (MASC) to take photographs of individual snow particles. This instrument,
representing a further advance in the field of imaging disdrometers, collects high-resolution stereoscopic photographs of
snowflakes in free fall while they cross the sampling area (Garrett et al., 2012), thus providing information about snowfall
microphysics (Praz et al., 2017). The optical structure of the imaging-disdrometer and the MASC makes these instruments
reliable in the presence of millimetre-sized hydrometeor precipitation. In Antarctica, their practical application is mostly
limited to coastal zones where particles are coarse (e.g. MASC resolution is 33 μm).
The direct observation of inner Antarctic particles requires imaging techniques with resolution of a few microns. Photographic
studies of precipitation in the interior of the Antarctic are quite rare, carried out primarily at the South Pole Station (SPS)
through formvar replicas. In early works with formvar, Hogan (1975) identified at SPS millimetre-sized columnar crystals and
column- and bullet-rosettes in cloud precipitation, and smaller ($\cong$100 μm diameter) platelike particles in clear-sky
precipitation. Satow (1983), working with formvar replicas on Mizuho plateau found prevalently single bullets and
combination of bullets. Long solid column crystals were also found (with an air temperature range from -42°C to -56°C) with
a mean length of 290 μm and a maximum length of 1.2 mm, with a mean aspect ratio of 18. Small (50-400 μm) hexagonal,
triangular, scalene and square plates were also observed. Kikuchi and Hogan (1979) collected formvar replicas of DD in the
summer at SPS, finding columnar crystals of 90 μm average lengths and plates as small as 50 μm in diameter. Ohtake and
Yogi (1979) classified winter ice crystal precipitation in Antarctica under six categories. These included large rosettes, bullets
and columns (millimetre-sized), thin hexagonal plates and columns (200 μm or less), and smaller crystals of various shapes
including triangular and polyhedral. Shimizu (1963) observed "long column" crystals in the winter at Byrd Station (80S,
120W). Size distributions of Antarctic DD in winter and spring were reported by Smiley et al. (1980) for particles larger than
50 um: they observed the same ice crystal forms that were reported earlier. Walden et al. (2003) studied DD, blowing snow,
and cloud precipitation in winter, at SPS, by collecting crystals on slides and analyzing them using microphotography. In their
study, columns with an average length of 60 μm and plates with an average diameter of 30 μm were found in DD. The direct
observation of ice precipitation on the plateau was typically carried out by means of formvar replicas and/or microphotography,
but these techniques take time, are difficult to implement throughout the year and are necessarily limited to short field
campaigns and samples of very limited size. Designing automatic instruments for the continuous, photographic study of
precipitation in such a harsh environment necessarily require several compromises between the high resolution of
microphotography and the robustness of outdoor optical instruments such as disdrometers. Lawson et al. (2006) worked at
SPS, in summer, using innovative Cloud Particle Imagers (CPIs), which replaced formvar replicas. This technique allowed the
automatic analysis of around 700,000 DD crystal images in terms of caliper size, aspect ratio and other shape parameters. An
automatic classification software, based on shape parameters, was used to categorize the images into nine simplified classes:
small plates and spheroids, columns, thick plates, plates, budding rosettes, rosettes, complex with side planes, irregulars.
Concordia International Station, located on the Dome-C (DC, 75°S,123°E, 3220 m above sea level) is a special location to test
new instruments for precipitation studies. Surface temperatures seldom exceed -25°C in summer, whereas winter temperatures
can reach -85°C. The 3 m average wind speed is 3 ms$^{-1}$ for Aristidi (2005) and 4.5
ms$^{-1}$ (hourly-averaged) for Argentini et al. (2014). The strongest winds (up to 15 ms$^{-1}$, hourly-averaged) blow from the

continental regions. These winds are due to gravity flows from the inner plateau regions south of Dome C, and are more often observed during the winter, especially in coincidence with warming events. The circulation at the surface during the summer is affected, especially in daytime, by the synoptic circulation. In summer the wind speed oscillates during the day, with values increasing (by a few $ms^{-1}$) in the afternoon, when a convective layer develops, leading to the increase of the wind speed (Argentini et al.,2014). Relative humidity relative to ice is typically around 55-85% (Genthon et al., 2022). In these conditions, precipitation of ice crystals can be studied by simply collecting them on horizontal surfaces. This is done at DC by hand, starting in 2008, collecting precipitation on flat surfaces ("benches") and visually inspecting it. This analysis is restricted to one observation per day, a rate that is difficult to increase, especially in winter. The analysis of these samples is also time-consuming and often subject to biases due to ice re-processing and sublimation, hoar formation, and subjective judgement of the shape and relative abundance of ice particles. Schlosser et. al (2017) relied on this manual observation and classification of ice particles in his analysis of precipitation isotope data at DC. They classified the ice grains into diamond dust, drifting snow, snow and frost (hoar). The prevalence of hoar in the observed daily precipitation record (with temperatures below -50°C) indicates the limitations of this manual technique if detailed information on DC precipitation particles is desired.

Detailed work was carried out in DC on a few individual DD and cloud precipitation crystal replicas by means of SEM electron microscopy by Santachiara et al. (2016). They also analyzed very small particles (10-50 μm), in a size range inaccessible to ordinary                                        optical                                        methods. The purpose of developing ICE-CAMERA was to fill a gap in precipitation monitoring at Concordia with a robust instrument capable of monitoring with continuity, all-year round, habit and size of ice particles in precipitation, while avoiding some of the problems associated with the visual inspection of precipitation. This was achieved through the combined development of robust camera equipment and machine learning techniques for sizing and classifying ice crystals.

## 2. The instrument.

### 2.1 Overview of ICE-CAMERA.

ICE-CAMERA is a flatbed scanner (Zheleznyak et al, 2015), whose operating principle is the same as that of ordinary flatbed scanners in offices. In the case of ICE-CAMERA, it specially designed for observing polar precipitation in the harsh environmental conditions of Concordia station (Fig. 1). Within this work, the term "precipitation" will include both "diamond dust" and cloud precipitation. The 'Deposition Surface' (DS) is defined as the horizontal, glass surface of the instrument facing the sky and collecting precipitation.

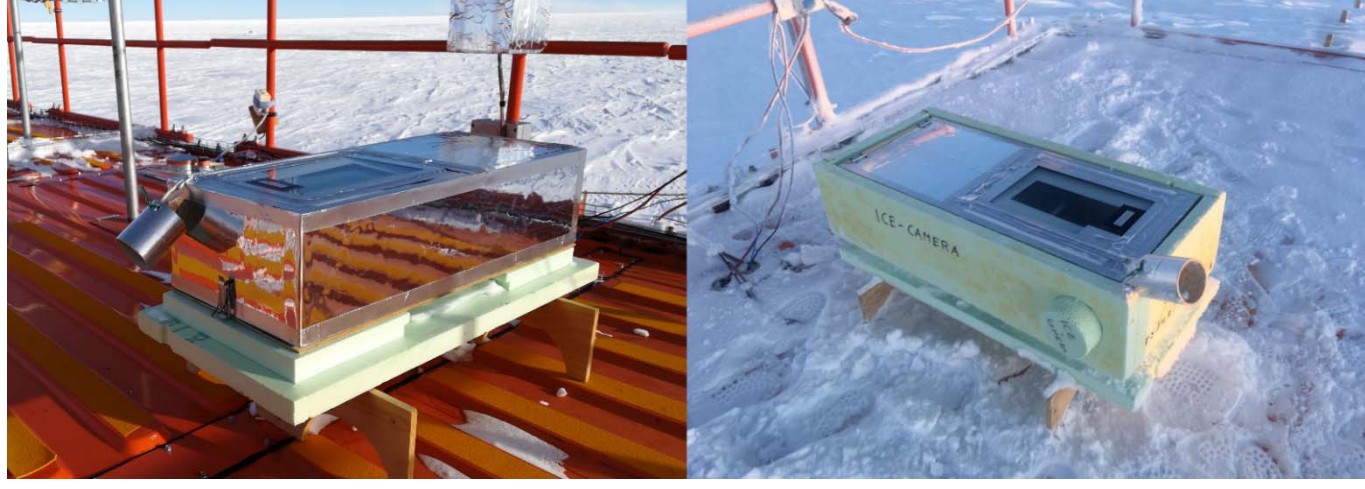

**Fig. 1: ICE-CAMERA with its summer sun-shield (left) and with the winter coat (right).**


The principle is simple: at the low temperatures and low wind speeds typically encountered at DC, precipitation falling on a
horizontal glass surface accumulates with time until it sublimates, leaving enough time for scanning the DS for counting and
measuring individual ice particles. The DS is the external surface of a special glass with electric heating (2.7). A second sheet
of glass together with the DS glass creates a double glass window that isolates the DS from the heated parts of the instrument
(Figs.2, 3). The scanning, like in ordinary flatbed scanners, is performed by means of a line-scan camera (Sect.2.2). moved by
a motorized scan sledge, and looking up at the DS through a 45° mirror (Fig. 2). The focus of the camera is adjusted by a
small motorized focusing sledge moving the 45° mirror (Sect.2.3). During the scan, the image is sent to the PC, located inside
the           shelter.
After a complete scan of the DS, the glass is heated and the precipitation sublimated (section 2.7). Once cooled down, the
clean SD begins to accumulate new particles. This cycle takes place every hour. After each image acquisition, the MATLAB
image processing code is called to process the DS image, and a summary-image containing only segmented particles (if
present) is stored for post-processing. (Sect. 4.1.3). Every particle is also automatically measured through image processing
(Sect. 4.1) and classified through machine learning (Sect.4.2). Individual particle data are stored in rows in a text file, along
with weather and housekeeping data, for post-processing and statistical analysis.


**Fig. 2: ICE-CAMERA basics: the scan sledge moves the image-acquisition line along the deposition surface. The focusing sledge**
**adjusts the focus.**

All basic operations of ICE-CAMERA, (with the exception of CAM acquisition) are driven by a custom microprocessor
(Microchip PIC) logic board (Fig. 3). The same PIC board reads the housekeeping temperature sensors (attached to the DS
and placed inside and outside the instrument), drives the stepper motors of the sledges as well as pumps and fans. The PIC
Board communicates with the main computer (located inside the shelter) through RS232. NI Labview software controls image
acquisition, reads maintenance data, and monitors PIC operations along the RS232 line. The line-scan camera communicates
with the PC via Gigabit Ethernet.
The instrument is placed outdoor, on the roof of the "Physique" shelter, approximately 6 m above the ground.
ICE-CAMERA was first installed in Concordia in 2012, but replaced in 2014 with its improved version, described here.
From then on, the instrument works year-round to produce precipitation data, every hour. Standard meteorological data are
automatically obtained from local weather station AWS MILOS 520.


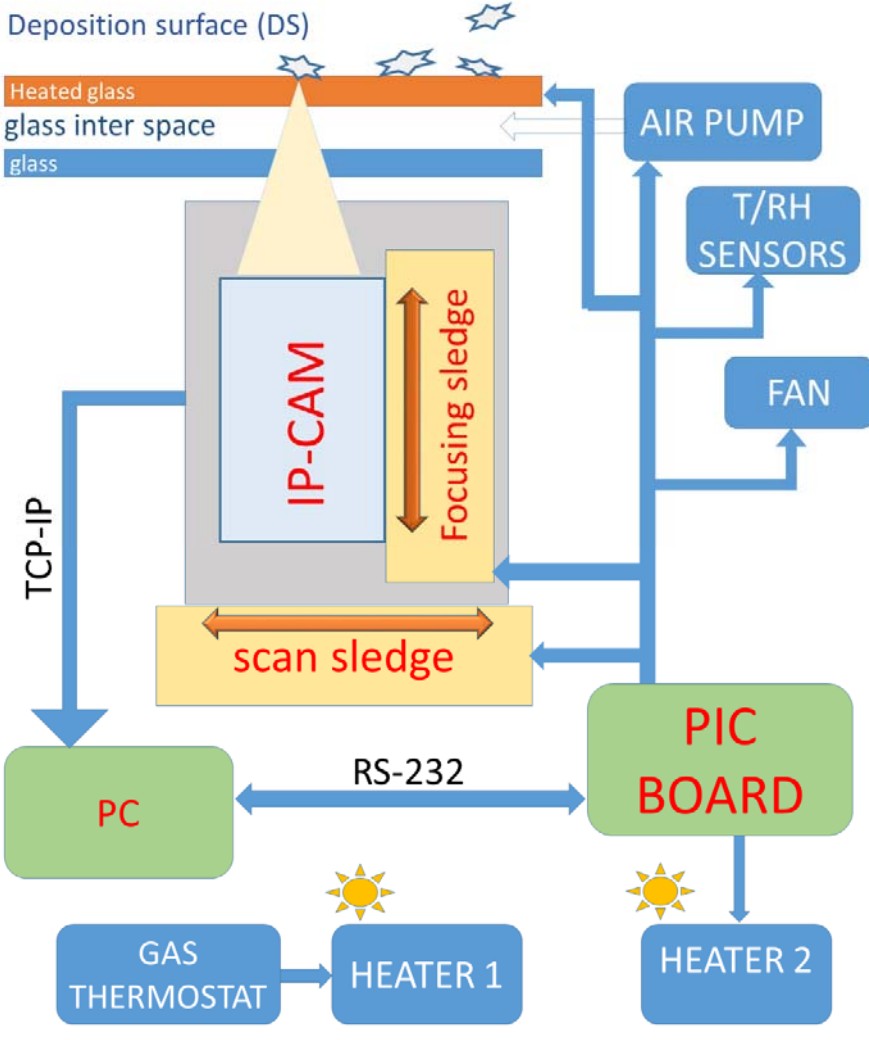



**Fig. 3: Basic schematics of the instrument**

**2.2 The line scan Camera.**
A linear scanning GigE Vision monochrome camera (Schafter-Kirchoff SK7500VTF-XB (52.5 mm sensor, 7500 pixels,
7x7μm pixels, 8.2kHz line frequency), equipped with a 1:1 macro lens (APO-Rodagon D1X, f5.6) is used for the acquisition.
The optics were designed by Schafter-Kirchoff  in order to have a resolution equivalent to the 7μm pixel size.  The 45 deg
mirror is used to look upward. The illumination is ensured by 850 nm LEDs. A colour filter (Schott RG715, 800-1000 nm
band-pass) was used on the CAM lens, in order to have a fully solar blind instrument. The line-scan camera assembly is moved,
hourly,  by a motorized sledge at a speed of 8 mm $s^{-1}$ in order to scan the rectangular DS ( 55x200 mm), located at the center
of the window.  The final image is 7500*30000 pixel, 12 bits, monochrome. A fine calibration of the actual pixel size of the
DS image was achieved by scanning a calibrated grid (0.1mm spacing) placed on the DS. This is necessary because the
effective resolution of the image produced by the moving linear camera along the sledge direction depends on how fast the
sledge moves. After the correction of this effect, the image pixel size resulted in 6.97 x 6.9 μm, which was extremely close to
the simulated size of 7x7 µm.. From the Nyquist sampling theorem, details less than 14 µm cannot be detected in the image
(under optimal focusing conditions). This resolution is enough, for example, for the observation of the hexagonal edges of the
smallest plates detected by the instrument.
**2.3 The Focusing.**
In working conditions, the focal depth is ±0.5 mm. A preliminary and accurate alignment of the motorized sledge plane to
the DS ensures uniformity of focus across the DS at air temperature. A motorized focusing sledge, moving the bending
mirror, allows to adjust the focus in operating conditions (Fig. 2). As ICE-CAMERA works outdoor at DC, it can
experience a broad internal temperature range, from +5°C in summer to -45°C in winter, with quite large temperature
gradients across the structure. Thermal expansion and changes in optical refractive indexes result in unpredictable changes in
the focal plane. The correction of the focus is thus automatically performed, every 6 hours, by bringing the measuring sledge
outside the DS, where a focusing spot (a sandpaper strip) is glued to the window (Fig. 4).

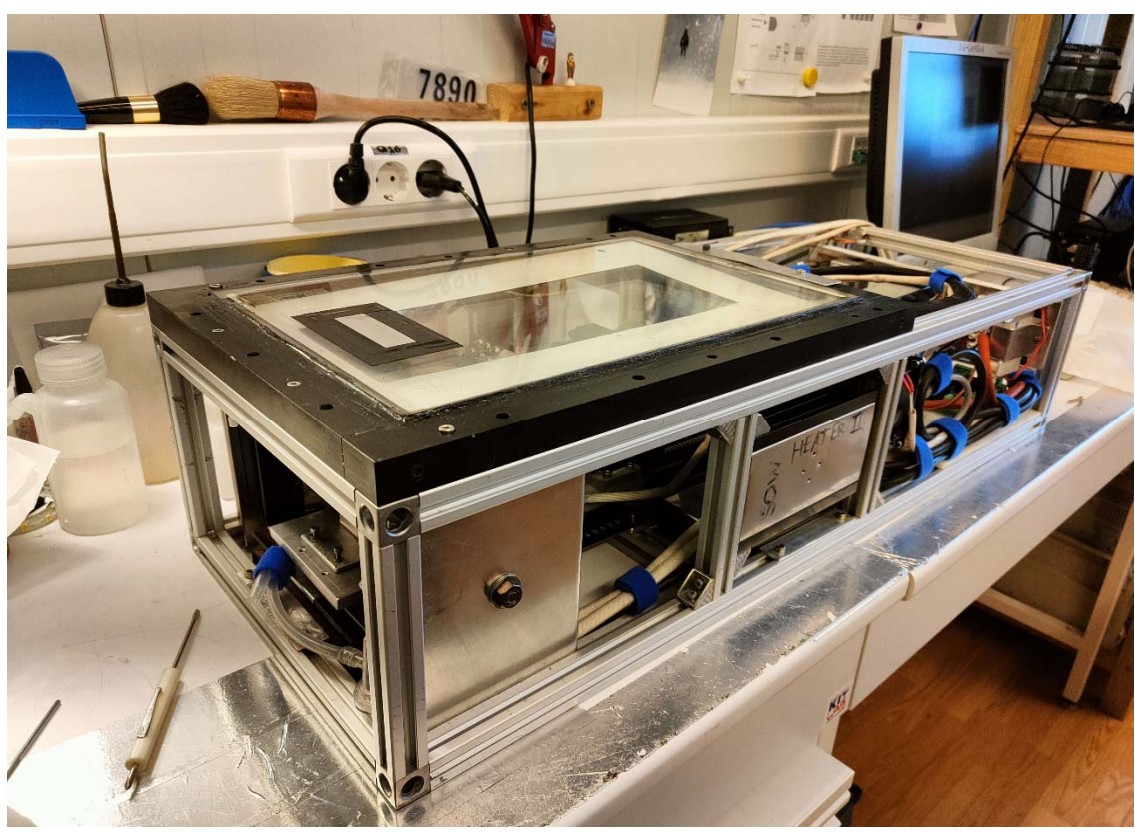


**Fig. 4: ICE-CAMERA out-of-the-box. The focus target if fixed onto the DS.**

The porous structure of the sandpaper has a length-scale of the order of 0.1 mm, comparable with the size of the measured
ice particles. While calibrating, the focusing sledge is moved by ±2 mm around the actual position in 0.25 mm steps.
Successive images of the sandpaper are taken and their contrast (defined as the standard deviation of the intensity of the
pixels) is measured. After a Gaussian-fit of the contrast as a function of defocusing (Fig. 5), the position corresponding to
the maximum contrast is obtained, and the mirror sledge is moved into that position. The typical focal spot adjustment
between two consecutive calibrations if 0-0.25 mm. The calibration takes approximately 5 minutes. For this reason, it is not
done after each measurement, so as to save PC resources for data processing.

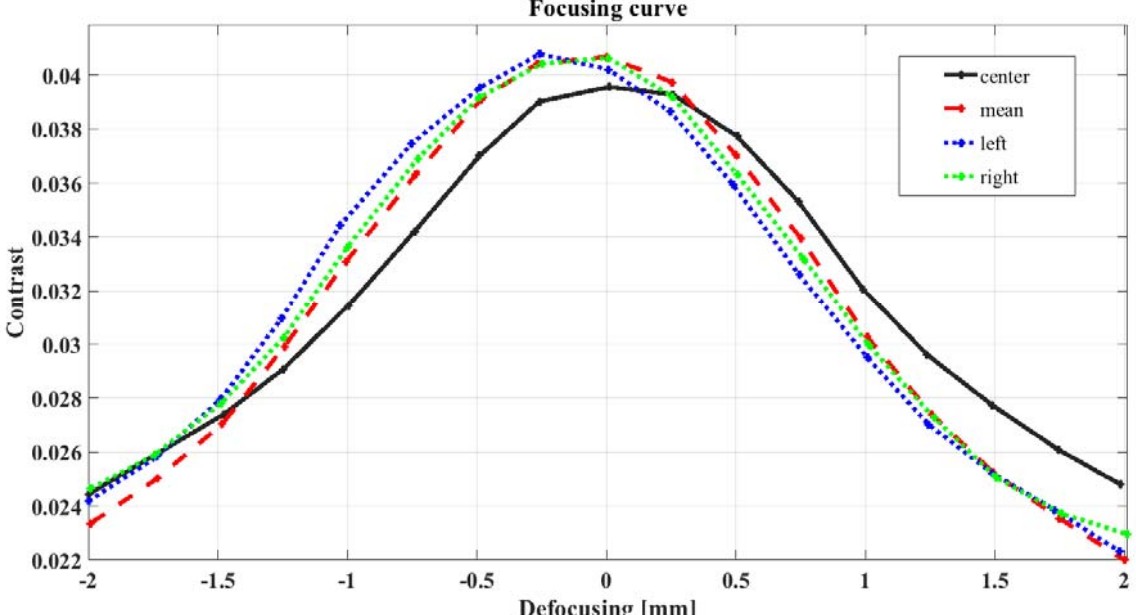



**Fig. 5: Typical focus calibration: Contrast is calculated in three sectors of the image: center, left and right. The contrast**

**throughout the image is also displayed (red). The slight difference in focus (0.2mm) between the center of the image and the side**

**wings is a normal lens effect.**

**2.4 Illumination.**

Lighting is supplied by two 850 nm LED (TSHG6200) strips. Both arrays illuminate the scan line symmetrically and approximately 45° from the optic axis in order to minimize multiple reflections in the double glass and within the camera lens. Infrared illumination was chosen in order to work in solar-blind conditions. This is particularly important, as the linear scanning camera always looks upward, to the sky. The uniformity of lighting along the linear CCD image was tested by taking an image of the same sandpaper used in the focus. The intensity profile along the CCD image was measured, and the intensity of the LEDs eventually changed to have a final intensity uniformity across the entire frame of less than 15%.

**2.5 The Deposition Surface (DS).**

The DS is the external surface of a 10 mm thick, electrically-heated glass (E-GLAS, Saint-Gobain). The glass is a sandwich with an electrically conductive layer pressed between two usual glass sheets. This glass is transparent at 850 nm, and can be electrically heated with 45 V ac, 95 W . A second, 2 mm thick, optically graded glass sheet (an ordinary flatbed scanner optical glass), placed 13 mm under the DS, makes up with the DS a double glass. This arrangement is necessary in order to keep the DS thermally insulated from the heated, interior of the instrument. A thermocouple is attached to the DS, while other thermocouples monitor double glass inter-space temperatures. A temperature of (at least) 3 °C above air temperature is enough to prevent, in DC, the formation of frost on surfaces in any season, as suggested by the work of Tremblin et al., (2011). In ICE-CAMERA, the temperature of the DS is usually 4 to 5°C above room temperature, which keeps the DS free of frost in all seasons (Fig. 6).

213

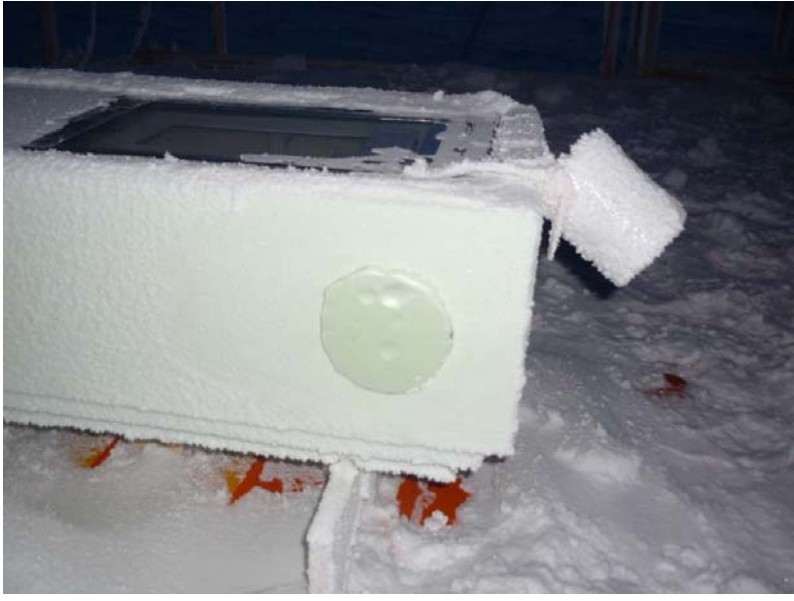

**Fig. 6: ICECAMERA at -70°C, Concordia station winter: the DS if free of frost**

During the sublimation period (Sect.2.7), ambient air is pumped for five minutes by means of a 3.5 l m$^{-1}$ miniature pump
through the double glass inter space, in order to keep the internal surfaces of the double glass always free of frost. Using
inert gases such as argon in the double glass space for the same purpose proved unsuccessful in Concordia at the extremely
low winter temperatures. In order to avoid the eventual accumulation of wind-drifted snow, the DS has no walls or obstacles
all around. Furthermore, the instrument is located on the roof of a shelter, almost 6 meters above the ground, an altitude
where blowing snow is not normally important at Concordia. Libois et al. (2014) identify drifting snow events at Dome C
when the 10-m wind speed exceeds 7 m s$^{-1}$. Assuming a logarithmic wind speed profile between the surface and 10-m and
an aerodynamic roughness length value of 1 mm (Vignon et al., 2016), this corresponds to a wind speed threshold value of 5
m s$^{-1}$ at 6 m above the ground. Winds below this threshold (near the annual average wind speed in DC) are not expected to
carry blowing snow to the DS. In addition, blowing snow impacts the flat horizontal and smooth DS at very small angles,
with a very limited chance of sticking to it. As a consequence, ice particles collected on the DS can be considered
representative of precipitation. In case of strong winds, not only the attachment of blowing snow to the DS is very low, but
also the collection of eventual precipitation is reduced. Since DS is warmer than air, there is no secondary growth in
deposited ice. Instead, the partial sublimation of ice particles before scanning could not be excluded, especially in summer.
This topic needs additional field work and will be modelled in Sect. 3.2.

**2.6 The thermal control.**
The temperatures measured by the ICE-CAMERA sensors are continuously transferred to the computer. The NI-Labview
software controls the internal temperature of ICE-CAMERA above -40°C (by driving the 200W, ventilated air heater "Heater
2" of Fig. 3), and the DS temperature always under -5° (by eventually disabling the "heated glass" of Fig.3). These conditions
are maintained throughout the year during every phase of the measuring cycle. An independent 200W thermostat ("Heater 1"
in Fig. 3) provides emergency temperature control in case of computer or PIC board failure. After a black-out, when the power
is restored, a timer is used to heat the inside of the instrument before turning on the electronics. This is important at Concordia
to prevent damage to standard electronics with typical operating temperatures of -40°C.
In winter, a 40 mm thick Styrofoam coat is added around the instrument for increasing thermal insulation, whereas in summer
a Mylar sunscreen prevents overheating of the instrument and allows keeping the DS below -5°C in the warmest days  (Fig.
1). Additionally, in warm weather, outdoor air is carried inside the box with a tangential fan, for better cooling of the
instrument.

**2.7 Sublimation-deposition cycle.**

After an entire  scan of the DS, electricity is applied to the DS glass to sublimate the particles. The heating rate of the DS
depends primarily on the electrical power applied to the glass and its thermal constant (approximately 0.8 W m$^{-1}$ K$^{-1}$ ) , and
secondarily on the wind speed. An indoor test (Fig.  7), showed a heating of rate of 2.5°C min$^{-1}$, and a cooling rate of 1 °C
min$^{-1}$.

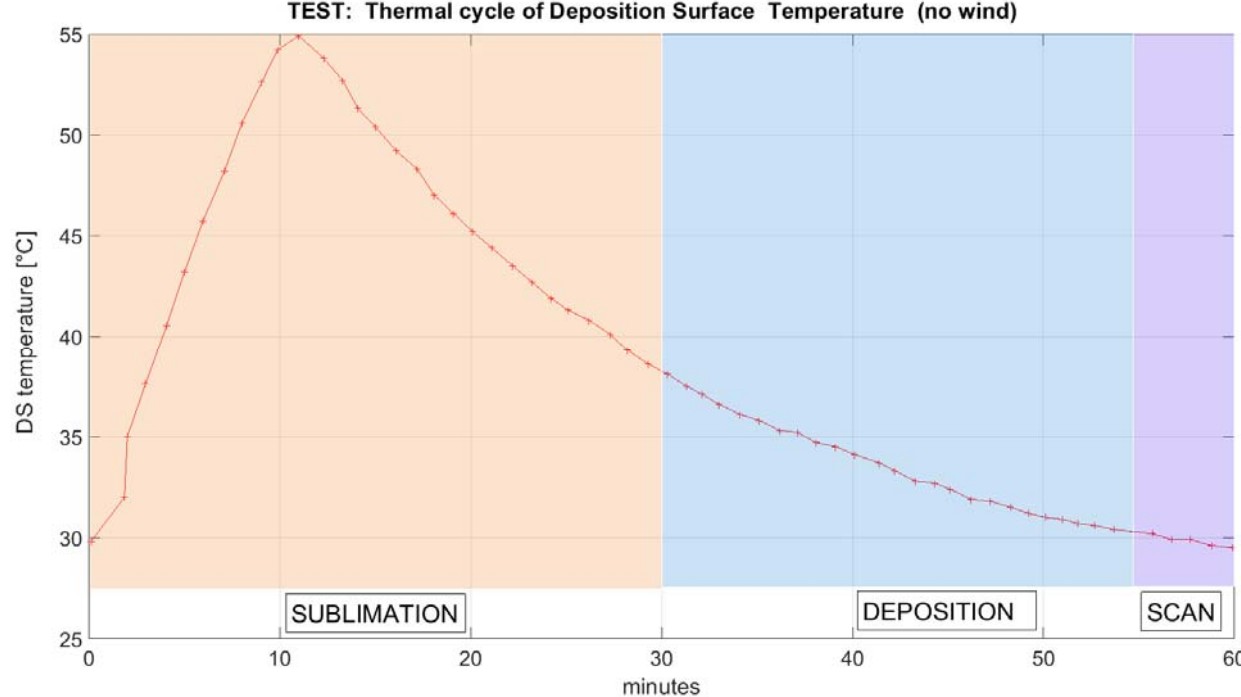

**Fig. 7 :  Indoor  test of DS heating-cooling within a 60 min cycle. For reasons of simplicity, the periods of sublimation and**
**deposition are well separated..**

The cooling rate is at most only about 50% of the heating rate. Cooling is passive through heat transfer to ambient air, with a
heat transfer coefficient of approximately k=0.024 W m-1 K$^{-1}$ in still air. k increases with radiation cooling, convection and
wind. During glass heating, heat is quickly transferred to the DS from the electrically heated inner layer, while during
cooling the heat transfers from the DS to the air occurs slowly, with a thermal constant k. This explains the asymmetrical
curve of Fig.7.
Outdoor tests carried out in summer at DC (-30°C air temperature) showed a heating rate of  3 °C min$^{-1}$ in still air, 2.5 °C
min$^{-1}$ with 2.5 m s$^{-1}$ wind speed, and 1.8 °C min$^{-1}$ with 5 m s$^{-1}$  wind speed. In all cases, the cooling rate was approximately
1.5 °C min$^{-1}$.
An outdoor sublimation test (-30°C air temperature, RH=60%, wind speed <3 m s$^{-1}$) performed with snow manually spread
on the DS showed that, after applying heating for 10 minutes (up to a DS temperature of  -8°C) the sublimation of the
majority of particles (diameter<1000 μm) was complete within 20 minutes after turning off the heating, with just a few big
grains (initial diameter>1000 μm)  still present after 30 minutes.
After these tests, the glass heating period was set at 10 minutes (the heating is stopped anyway if the DS temperature exceeds
-5°C to avoid melting of the ice in summer). At the peak of the sublimation period, DS resulted warmer than air of about dT
= 20°C.  Once the heater is turned off, and after a cooling time of approximately 20 minutes, the DS temperature comes back
to be warmer than the air by only 4-5°C. At this point the "sublimation period" (of approximately 30 minutes) is considered
complete, and ice particles start accumulating again on the DS, with no relevant sublimation, i.e. the "deposition period"
begins (as sketched in fig.7 for the indoor test). At the end of the deposition period, a scan of the DS is carried out, for a
duration of one minute. If no ice particles were detected on the previous scan, the DS heater is not applied, sublimation is not
needed. The effective deposition period depends on the temperature, wind and exposure to the sun in summer. This
uncertainty, combined with occasional wind removal and particulate sublimation (Sect. 3.2) during the deposition period,
prevents the use of ICE-CAMERA for rigorous quantitative precipitation studies.
DS surface temperature is actually measured by using a small thermocouple. This measurement implies great uncertainties due
to the radiant warming of the sensor in summer and the difficult thermal coupling with the glass surface. A non-contact
measurement of DS temperature by means of IR sensors would also be uneffective in winter conditions.

**3. Ice particles and the deposition surface.**
**3.1 Adhesion of ice particles on the DS.**
The adhesion of ice crystals to the smooth DS is caused by two principal reasons: Van der Waals and electrostatic forces.
Eidevåg et al (2020) studied the adhesion of dry snow particles after 90° impact to different wall materials (gravity is a minor
force in this application, and therefore their work applies to any wall orientation). They considered models for normal direction,
tangential sliding, and tangential rolling that account for the adhesive interaction of spherical ice particles (25-275 μm
diameter) and their aggregates. The Johnson-Kendall-Roberts (JKR) model for adhesion was used. Their findings showed that
the maximum normal velocity at which spherical ice particles adhere to a glass surface (critical stick velocity) decreases with
decreasing particle diameter. Spherical particles of 100 μm would adhere for speeds less than 0.02 m s$^{-1}$. The numerical method
of Eidevåg et al (2020) has been applied in the present work to compute the critical sticking velocity of spherical ice particles
up to 1000 μm diameter. The sedimentation velocity of ice crystals up to 1000 μm was also calculated using the formulation
of Böhm (1989)  for  DC conditions (Tair=-50°C ,  air density=1.03 kg m$^{-3}$, dynamic viscosity= 1.45E-5 Pa s). Particles
impacting on the DS with a sedimentation velocity lower than the critical sticking velocity are immediately captured with
100% efficiency by Van der Waals forces. The simulation shows that only spherical particles smaller than 50 μm  in diameter
fall on the DS with a speed smaller than the critical stick velocity, and thus stick immediately. This result does not change with
the different forms of particles, as for these sizes the speed of sedimentation is close to that of Stokes.  Particles above 50 μm
in diameter have an excess of kinetic energy to bind effectively immediately on the DS. However, the impact surface (DS) is
horizontal, so that the excess kinetic energy is rapidly dissipated in one or more vertical rebounds, until the critical sticking
velocity is achieved (Chokshi et al.,1993). So the adherence of ice crystals to the DS could also be explained by the forces of
Van der Waals alone. Ryzhkin and Petrenko (1997)  showed that static charges, naturally transported by ice crystals, increase
adhesion. The electrostatic interaction between the ice and the surface is significantly stronger than the van der Waals forces
at distances greater than the inter-molecular forces. Electrostatic forces are therefore expected to significantly improve the
adhesion of large ice particles to the DS. Once attached to the DS, the weak winds generally observed at DC cannot detach the
particles from the DS. Particulates are protected by the boundary layer (BL) that forms on the DS. The 99% thickness of the
laminar BL (Blasius solution) at the centre of the DS (0.15 m distance from the glass edge) is expected to be 7 mm at -50°C
with a wind speed of 1 m s$^{-1}$, decreasing to 2 mm at 10 m s$^{-1}$. As a result, the particles deposited on the DS are protected against
the wind.




**3.2 Sublimation of ice particles.**
The DS is always warmer than the surrounding air. This is necessary to eliminate hoar, enabling the device to be used in all
DC conditions. The undesirable effect is the accelerated natural sublimation of deposited particles. A wide range of
experimental and theoretical research efforts has characterized the effects of temperature and super-saturation on ice crystal
growth rates and morphology under conditions relevant to atmospheric processes (for example Lamb and Hobbs, 1971;
Libbrecht, 2005; Libbrecht, 2017). The wide variety of ice crystals found in nature has sparked an interest. Sublimation was
sometimes regarded either as the opposite process, or a less intriguing process, and was less visited in lab studies.
Nelson,(1988) sublimated numerous, 100 μm diameter plate crystals ( 0.1°C>T>-18°C, 0.05% to 5% sub-saturation) showing
that the crystals first lost sharp edges, and finally evolved into spheroidal particles, and the aspect ratio remained almost
constant. The sublimation rates were accurately predicted by the diffusion equation with the surface vapour density at the
equilibrium value for a uniform surface temperature. The sublimating crystal reaches a self-preserving shape that is one of the
shape preserving solutions of the diffusion equation.  Ham (1959) showed that ellipsoids and thus spheroids preserve shape
during growth and sublimation if the grain surface has a uniform temperature. Jambon-Puillet et al.(2018) also showed
experimentally and theoretically that sublimation first smooths out regions of sharp curvature, leading to an ellipsoid. The
second stage is the sublimation of the self preserved ellipsoid shape. The entire process may be modelled as a vapour diffusion
problem, mathematically equivalent to the resolution of the electrical potential around a charged conductor. Using this analogy,
they provided a mathematical method for simulating the sublimation of the ice particle. The sublimation of the ellipsoid turned
out mathematically simple, and their method was adopted in this work to numerically simulate the second stage of sublimation
of ICE-CAMERA particles.
Monodispersed oblate spheroids with an aspect ratio (AR) of 5, in thermal equilibrium with the DS, were assumed in the
simulations as a surrogate for ice plates. The two major spheroid axes coincide with the «diameter» of the oblate spheroid, D.
In the model, D, DS temperature, air temperature and relative humidity with respect to ice (RHair) can be changed. The
sublimation time required for full sublimation of a spheroidal ice particle was computed. As sublimation accelerates when the
particle is going to vanish, the time necessary for the complete sublimation is only slightly larger than the time necessary to
reduce the particle to the minimum particle size (D=60 μm) accepted by ICE-CAMERA image processing. The simulations
assume that the preliminary sublimation of the high-curvature parts of the particle (sharp edges, corners, surface irregularities)
was already completed, so that the calculated  time of sublimation must be considered as a lower limit for real-world crystals,
and  probably almost one half of the overall duration of sublimation (Jambon-Puillet et al.,2018). Simulations also assume the
thermal equilibrium between the particle and DS, a condition which is not necessarily satisfied on the thermally insulating
glass surface of the DS. Figure 8a shows the total sublimation time with the DS heated dT=+20°C above air temperature
(sublimation period). The humidity resulted irrelevant in this case, and only results for 70% RHair  are shown. Results show
that at -30°C air temperature (summer conditions in DC) complete sublimation can occur within a few minutes after attaining
the DS sublimation temperature, for all particle sizes up to 1 mm. At lower air temperatures, the sublimation time increases:
at -70 °C (winter temperature in DC), particles smaller than 100 μm in diameter still disappear within 10 minutes, while larger
particles can survive along the sublimation period. Simulations showed that, at -70 °C, dT should be increased to dT=60°C in
order to ensure the complete sublimation of ice particles up to 1 mm diameter during the sublimation period. This is actually
not possible with the electric heated glass adopted, but could probably be achieved by microwave heating.

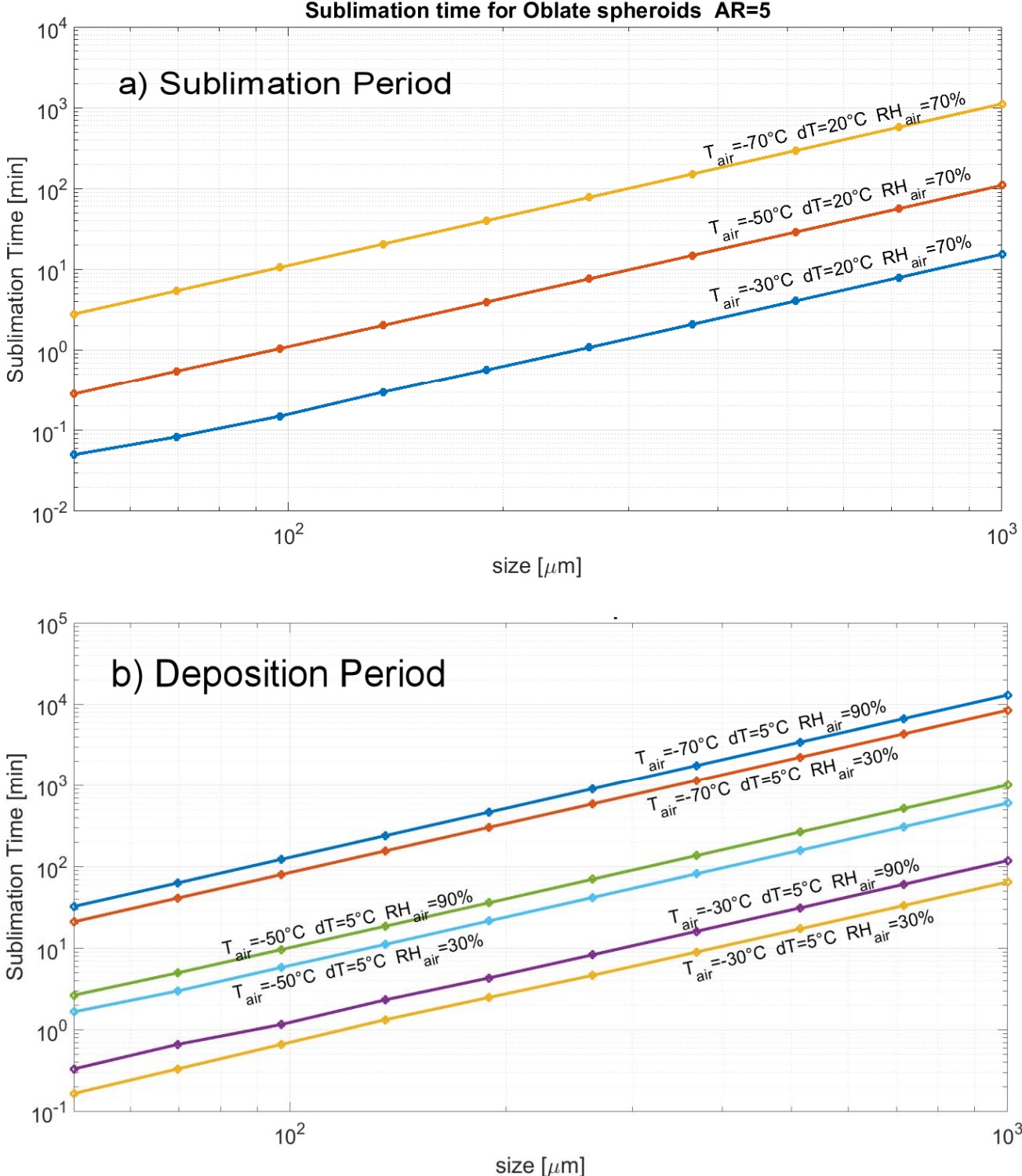

**Fig. 8: Sublimation time of monodisperse oblate spheroids at varying air temperatures, with a) dT = 20°C (sublimation period) and b) dT=5°C (deposition period).**

After the sublimation period, most of the particles previously collected on the DS are sublimated, and a new deposition period begins. Even during this period, sublimation still acts on ice particles, albeit slowly. Figure 8b shows the sublimation time expected for monodisperse spheroids during the deposition period. The DS was considered 5 °C hotter than air. As shown, during the deposition period the relative humidity of air also plays a role, even if secondary. In summer ($T_{air}$=-30°C), sublimation can take less than a minute for particles smaller than 100 μm and ten minutes for 300 μm particles. During winter ($T_{air}$ = -70°C), all particles are expected to survive through the deposition period. As a rule-of-the-thumb, simulation showed

that working with dT=+5°C resulted in an increase of the rate of sublimation by a factor 2-3 compared with a DS in thermal
equilibrium with ambient air (dT=0) for the whole range of air temperatures and RHair shown in Fig. 8b.
Results of Fig.8b show that the effective lower limit of ICE-CAMERA particle detection is not limited solely to the resolution
of the optical system and/or image processing software. In summer, particles smaller than 100 μm may be decimated during
deposition, unless they fall just before scanning. Such small particles dominate diamond dust events. As a result, ICE-
CAMERA, during the summer period, is best suited to the study of cloud precipitation. Nevertheless, visual screening of ICE-
CAMERA images showed only a limited number of small particles revealing signs of partial sublimation, such as rounded
corners, smooth edges, or a spheroidal appearance. Some small plates (observed mainly in winter, when sublimation during
the deposition period is very slow) showed smoothed corners, but it is not clear if this was induced by sublimation or is a
natural feature of these ice grains. Also, even in summer, small DD particles such as plates (with no signs of edge smoothing)
were normally observed (Sec.5.2). It is probable that most particles (other than, probably, pristine plates) never achieve thermal
equilibrium with the DS glass, and that the results of Fig.8 should be considered as the worst case. Also, the sublimation of
the high-curvature parts of the particle prior to assuming the spheroidal form (Jambon-Puillet et al.,2018) could take much
more time than the sublimation time calculated here for the spheroid. A series of consecutive DS scans at fixed air temperatures
is needed to measure the effective sublimation rate of small particles in deposition conditions (dT=+5°C).
When a polydisperse particle population is deposited on the DS instead of monodisperse particles, a more complicated
sublimation picture arises, because small spheroidal particles, shrinking, are continuously replaced in the size distribution by
sublimating, initially bigger ones. An initial uniform particle size distribution (PSD) of the oblate spheroids (AR=5) was
assumed with diameters between D=1 and 2000 μm for the simulations. The evolution over time (1 sec resolution) of the PSD
was calculated (Fig. 9) in terms of particle survival (the ratio between the actual number of particles in a certain size bin and
the initial number in the same bin). No vapour competition between ice particles was taken into consideration in the
simulations. Results are similar to those of monodisperse particles (Fig. 8), with a slightly longer time of sublimation for
polydisperse particles compared to monodisperse particles of the same size. Results for an air temperature of -70°C confirm
that most particles larger than 500 μm survive, throughout the DS sublimation period (dT = 20°C), longer than 30 minutes.
This means that sublimating by heating the glass is quite inefficient for large particles in winter. During the deposition period,
at -70°C losses for sublimation are scarce and limited to particles smaller than 200 μm. Consequently, double counting of the
same particle (D>500 μm) is possible in two consecutive ICE-CAMERA scans in the cold DC winter. At -30°C air temperature
(summer) the heating of the DS with dT=20°C leads to the sublimation of most particles up to 2 mm diameter within 5 minutes.
On the other side, during the deposition period particles smaller than 500 μm can undergo sublimation over a period of just 10
minutes in summer, thus limiting the effective period of deposition before a scan. As with monodisperse particles, this
introduces bias in the summer because many small particles (typical of DD) can be removed before they are measured.

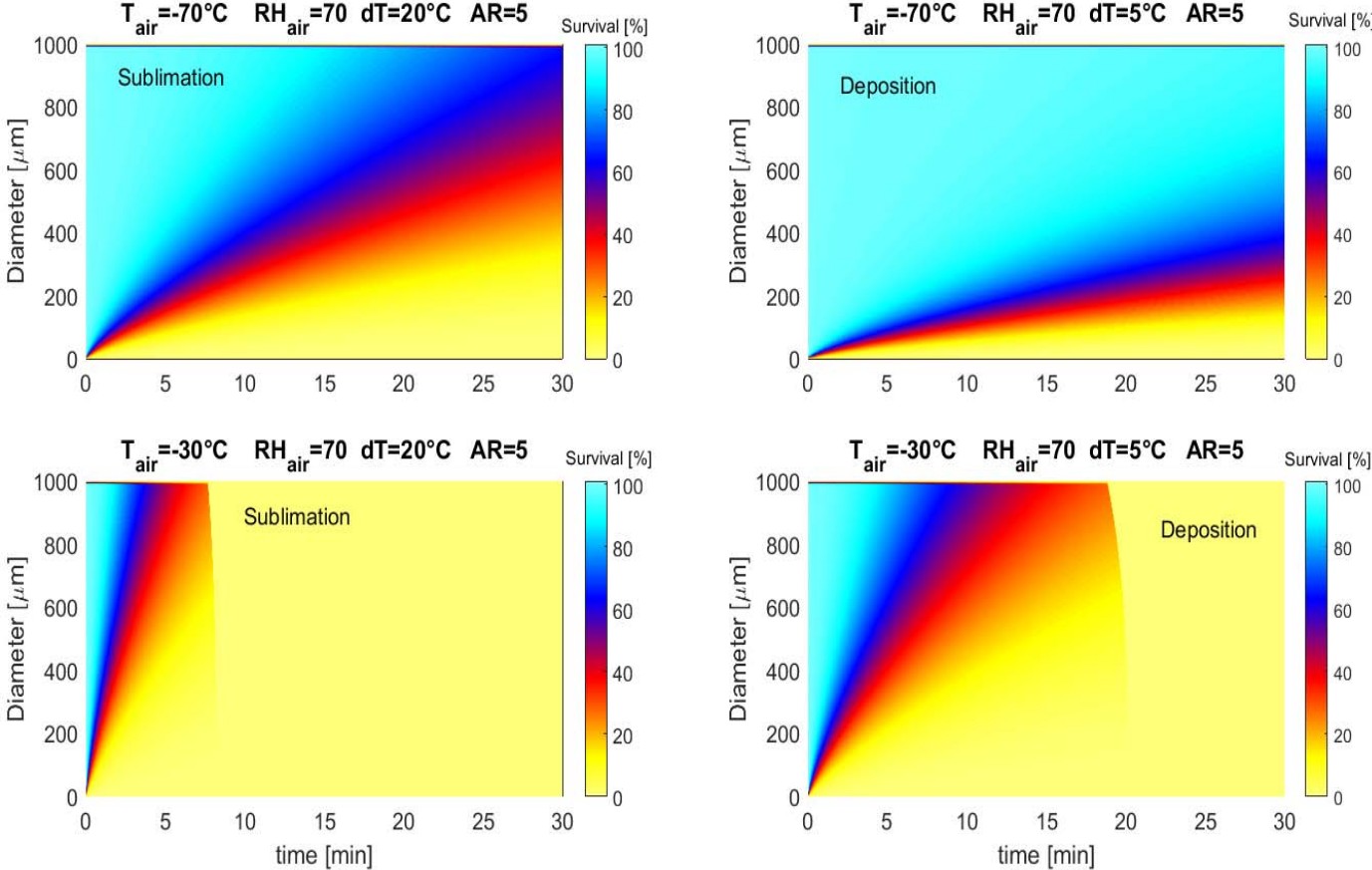

**Fig.9: Evolution of a originally uniform PSD of ice spheroids (D=0-2000 μm, AR=5) under different atmospheric conditions. (RHair is a secondary factor affecting the results, shown here for RH=70%). Left: sublimation period; right: deposition period. Top: winter, bottom: summer.**

Even if these results could be disappointing for interpreting ICE-CAMERA data, the same problems affect the actual method of observing precipitation in DC: collecting and observing (every 24 hours) the ice particles deposited on flat surfaces ('benches') is affected by the same problem as collecting particles on the ICE-CAMERA DS with dT=0. Fluctuations in relative humidity over 24 hours result in sublimation and regrowth of particles on the "benches" in an almost unpredictable manner. Fig. 10 shows the expected sublimation time for particles (with the same PSD of Fig. 9) placed on 'benches' (or ICE-CAMERA DS) in equilibrium with air (dT=0) for extreme, sub-saturated conditions: winter $T_{air}$=-70° ($RH_{air}$=30% and 99%), and summer $T_{air}$=-30 ($RH_{air}$=30% and 99%). The PSD evolution is computed with a resolution of 1 sec for a total period of 6 hours. The results show that sublimation also works in winter and with almost saturated air (99% $RH_{air}$), leading to a complete loss of small particles (D<200 μm) in a few hours. In summer conditions and 30% $RH_{air}$ sublimation happens much more quickly, with the disappearance of all particles up to 2000 μm in 30 minutes. With $RH_{air}$=99%, sublimation removes all particles in just a few hours in summer. In presence of wind and dry air, sublimation rate could even increase, as observed by Grazioli et al., 2017 in coastal areas. These simulations all refer to sub-saturated conditions: in the case of a 'bench' in thermal equilibrium with super-saturated air, hoar form on the surface, with a possible confusion with precipitation.

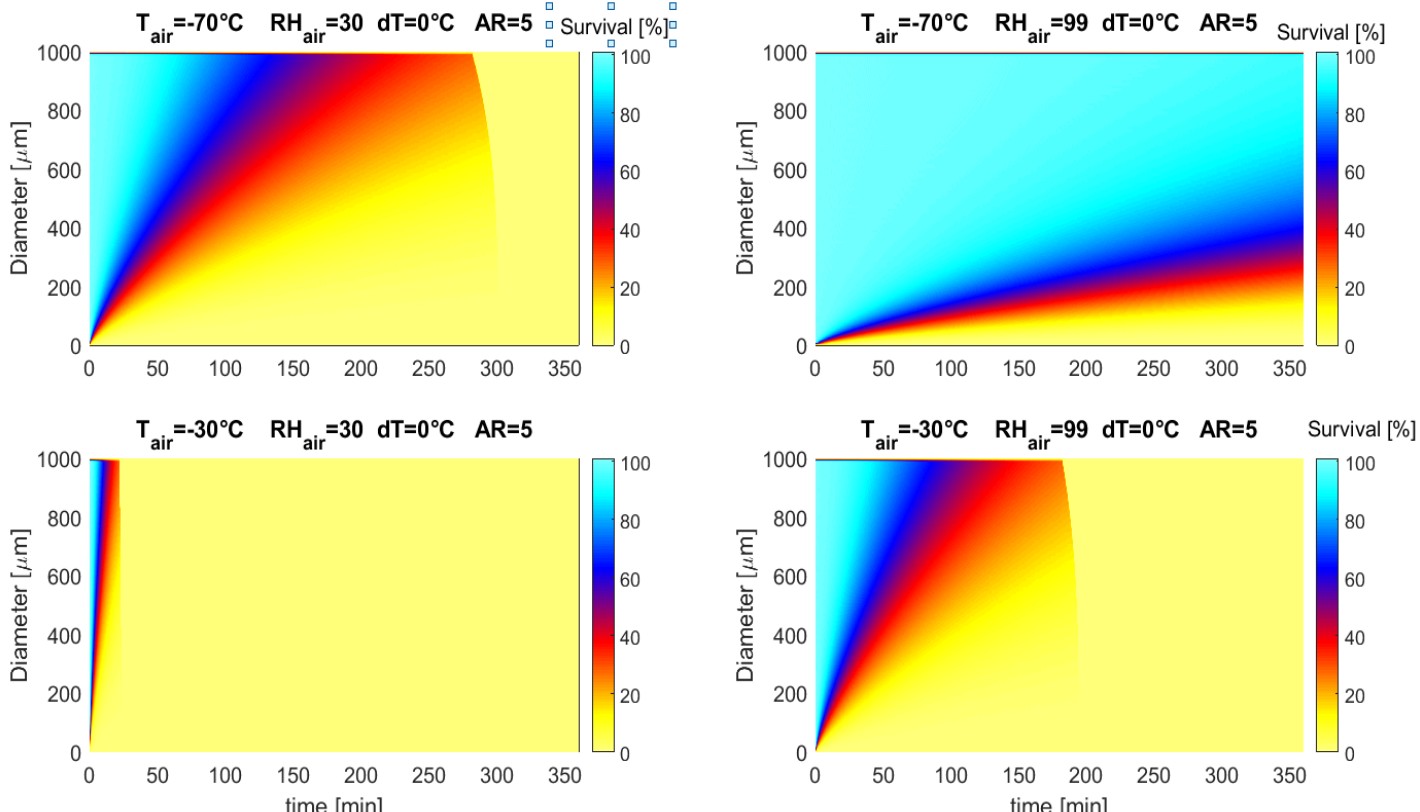


**Fig.10: Evolution of a originally uniform PSD of ice spheroids (D=0-2000 μm, AR=5) under different atmospheric**
**conditions. The DS (or 'bench') is in thermal equilibrium with air (dT=0). Top: winter,  bottom: summer**


**4. Data processing.**

**4.1 Image processing.**
ICE-CAMERA is not just designed to take photographs of ice particles, but to provide automatic morphometry and
classification of polar precipitation. This was accomplished through the use of image processing and machine learning
techniques. The process is divided into two parts: segmentation and measuring, and classification of ice crystals.

**4.1.1 Image segmentation and measurement of ice particles.**
After acquisition, the raw ICE-CAMERA scans are segmented, using MATLAB software, to isolate all detected particles. The
process follows the workflow of Fig.  11.  Refer to Pratt (2007) for image-processing nomenclature, to Walton, 1948 for Feret
measurement, to Russ and Brent Neal (2017) for the nomenclature of standard shape parameters such as Eccentricity, Euler
Number, circularity, roundness, solidity, compactness, form factor, and number of skeletal branches. The normalized central
moments f1...f7 were also computed as described by Hu, (1962).
The Aspect Ratio (AR) is defined as Feret's length/ Feret's width. The Feret-box surface-equivalent diameter (Df) is defined
as the diameter of the circle of the same area as the Feret bounding box, while the  surface-equivalent diameter (Ds) is
defined as the diameter of the circle having the same area as the segmented ice grain. The main steps of  Fig. 11 are visually
summarized in Fig.12 for a rimed, columnar particle.



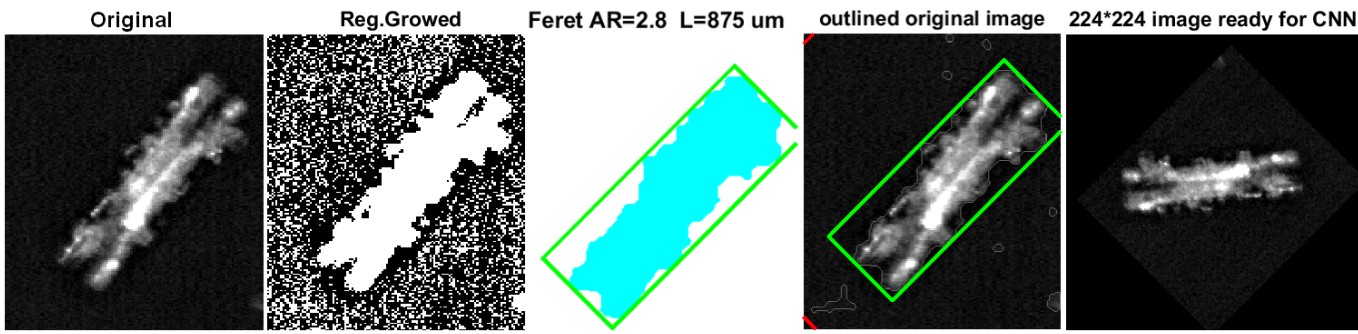

**Scan** → background subtraction → intensity equalization → Region growing (with normalized-intensity range: 0.2-1 )

Image Segmentation and Labeling (8 pixel -connected regions) ← "majority" correction (all pixels surrounded by 4 other "one" pixels are set to "one"). ← Image closure (10 pixels)

Selection of the largest labeled region → Creating the best adapted, rotated Feret bounding box → Image rotation (horizontal box) → Image resize to CNN input size 224*224

Feret's length, width, Aspect Ratio

Feret box surface-equivalent diameter

Standard measurements :

projected surface (S)

surface-equivalent diameter (Ds).

shape parameters

(Eccentricity, Euler Number, circularity, roundness, solidity, compactness, form factor, skeletal branches)

**Fig. 11: The image-processing flow chart.**

| Original | Reg.Growed | Feret AR=2.8 L=875 um | outlined original image | 224*224 image ready for CNN |

**Fig. 12: The original image (in this case a rimed column) is segmented using 'region-growing'. The projected particle area (clear blue) is calculated. The bounding box is determined (green) and the Feret length and width measured. The image is finally rotated to have the mayor axis horizontal, re-scaled, and resized to the CNN input size.**

### 4.1.2 Summary-image of detected particles.

The bounding boxes of all individual ice particles detected in a scan are sorted by Feret length, and reassembled in a summary-image collecting all segmented particles (Fig 13). Each particle is also associated with a numerical record containing the coordinates of its bounding rectangle on the summary-image, shape parameters, time of acquisition and local weather data. In this way, the re-analysis of the summary-image is possible instead of re-processing the original, large image. The original image is ultimately removed.

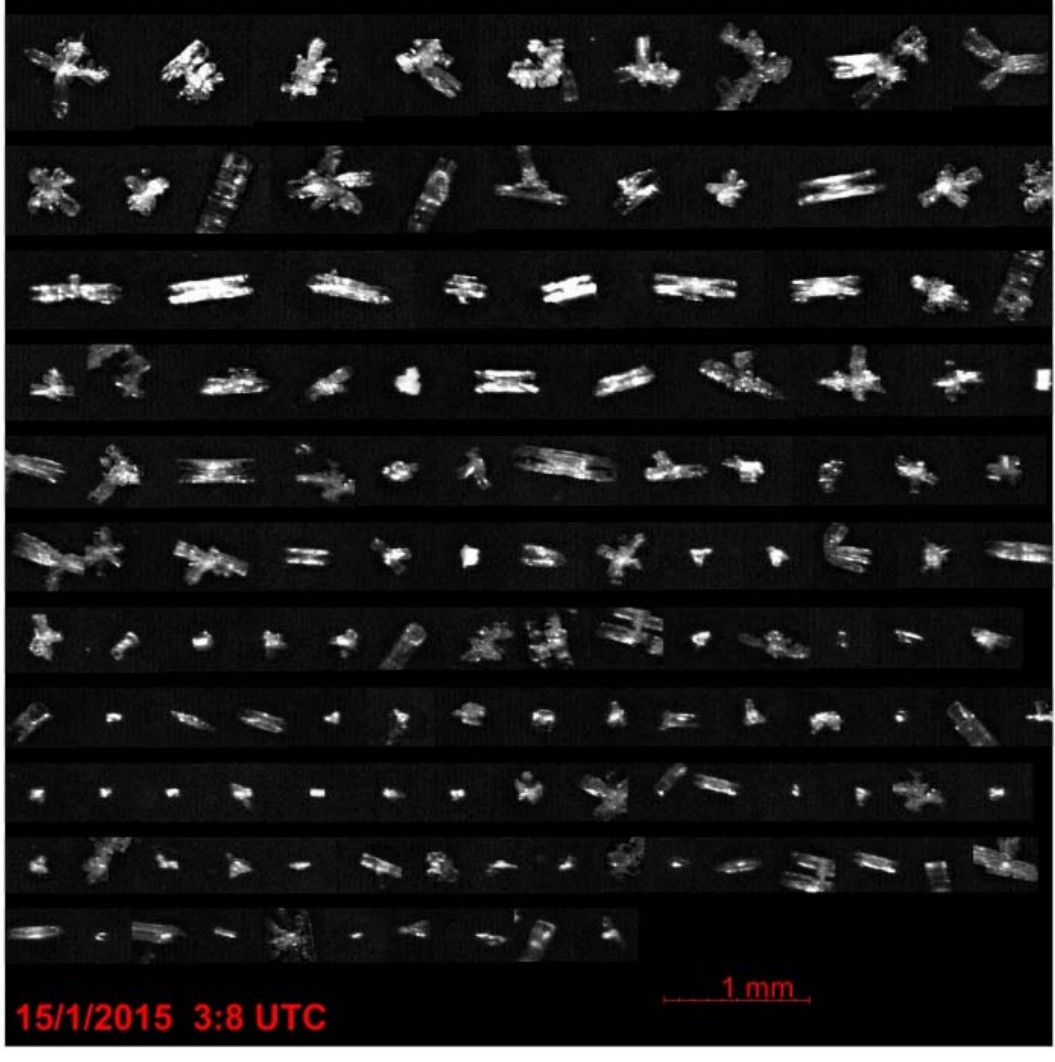

**Fig. 13: Example of a summary-image for a single scan.**

### 4.1.3 Limitations and uncertainties in detecting and sizing ice particles.

1) The total number of particles measured is actually limited to 2000 per scan, as a result of MATLAB memory limitations. Extra particles are not treated.

2) Particles below 3600 μm² in bounding-box surface, (equivalent to approximately 60 μm in diameter for a spherical particle) are not preprocessed (smaller particles could be detected, but most have a seemingly circular shape due to low pixelation or poor focus).

3) The segmentation becomes difficult when overlapping particles or aggregates of particles are present. In such situations, double counting of the same particle may occur in up to 12% in a scan in the presence of an intense precipitation event. The same particles can in fact fall inside different segmented areas of the image, because of the

lack, on the original image, of defined boundaries between particles. The process of "region growing" which leads
to segmented particles can actually start, independently, from several bright ("seed") regions located in different parts
of the image of the overlapping particles. The 'region growing' processes can then propagate through the overlapping
particles leading to several 'copies' of the same, segmented image. This unwanted effect could be prevented by
looking for similar copies of the same segmented image, but this method was not implemented at DC due to limited
PC resources. Overlapping particles are normally classed by the CNN algorithm as "clusters". A few occasional
arrangements of three or more overlapping columns are sometimes mistaken for single plates. The Feret measurement
of these particles is meaningless. At DC this situation occurs only after heavy cloud precipitation, a relatively rare
event.

4)    Multiple counts of the same particle also occur for non overlapping particles when multiple bright spots exist within
the same particle. As in 3), the region-growing process can start independently from several 'hotspots', leading to
false copies of the same segmented particle. This effect could potentially be avoided by comparing segmented images
and deleting copies, but this method was not implemented at DC.

5)    Particles close each other in the original image could be segmented into a single particle by region-growing and thus
misclassified.

6)    In the case of defocused images, the particle shapes are all close to a fuzzy, round or elliptical shape, which can cause
a misclassification into irregular particles, spheroidal particles or plates. ICE-CAMERA images dominated by this
type of particles are normally eliminated during a preliminary manual screening. Also, a few big particles in summer
resulted rounded by partial sublimation. A few images containing only rounded or "spheroidal" particles of 500 μm
diameter or greater were collected during the warmest part of summer, and were manually discarded before the
statistical data analysis.

7)    Needles and hexagonal plates (typically small, see Fig.23) may be very bright in ICE-CAMERA images due to
enhanced light diffusion at preferred angles. For the same reason, hollow columns sometimes have a shiny spot in the
middle. .In the case of needles, this effect can reduce the apparent aspect ratio, as the width is apparently increased
by the scattered light saturating the camera. For plates, the bright specular reflection blurs sometimes the polygonal
contour, especially in the case of small plates.


## 4.2 Automated classification of ice particles.

An initial attempt at automatic classification of ICE-CAMERA segmented images was made in 2014 using shape factors. This
kind of technique has also been used by others (e.g. Lindqvist et al., 2012) for attempting the classification of ice particles. In
the case of ICE-CAMERA this approach resulted extremely unreliable. A much more promising approach was offered after
2015 by the rapid development of transfer learning and convolutional neural networks (CNN) (Le Cun et al., 2015;
Schmidhuber, 2014). Xiao et al. (2019) successfully applied deep transfer learning to ice particle images obtained with airborne
Cloud Particle Imagers (CPI). The CNN approach has added much value to ICE-CAMERA because a reliable classification
of ice particles into simplified classes became possible. The CNN used for the ICE-CAMERA particle classification is
"GoogleNet" (Szegedy et al. 2015), a variant of the Inception network, a deep convolutional neuronal network developed by
Google scientists. GoogleNet is a type of convolutional neural network based on the Inception architecture. It utilises Inception
modules, which allow the network to choose between multiple convolutional filter sizes in each block. The GoogleNet
architecture consists of 22 layers (27 layers including pooling layers), and part of these layers are a total of 9 inception modules.
In this work, GoogleNet was used in MATLAB R2020b environment. The GoogleNet CNN, pretrained on the ImageNet
dataset (Deng et al. 2009), was used, with its final, fully connected layer changed to size 14. The input layer of the GoogleNet
architecture requires images of size 224 x 224.

**4.2.1 The CNN classification classes.**

Low temperatures and humidity on the high Antarctic plateau reduce the diversity of ice particle shapes. This is observed on

the field at DC, at South Pole station (Lawson et al., 2006), and suggested by review works such as Bailey and Hallett (2009).

Following an initial survey of the ICE-CAMERA image database, a set of 14 types of particles was selected, as shown in Fig.

14. When choosing the 14 classes, I assumed that shapes easily recognizable by a human operator could also be easily

recognizable by a CNN.

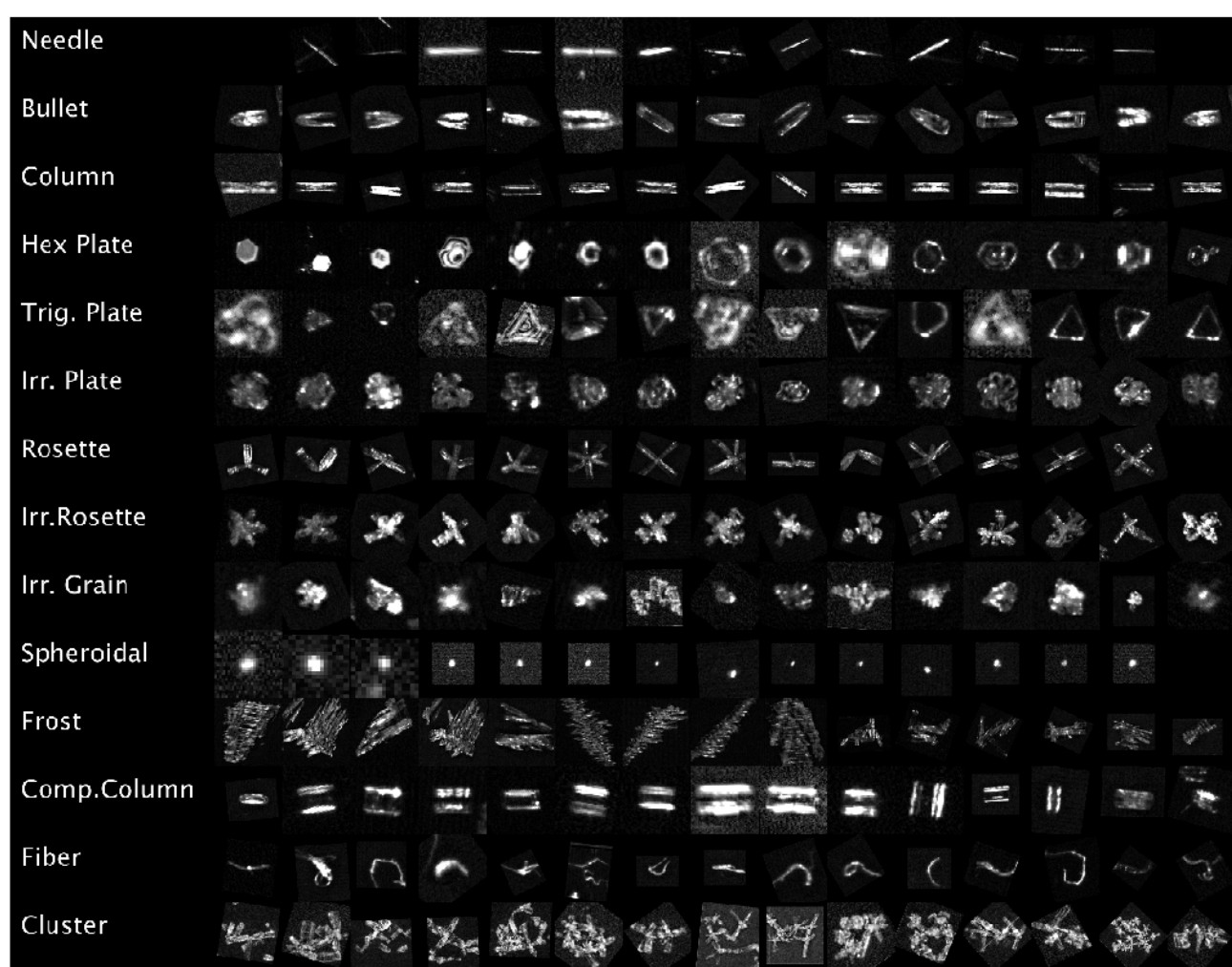

Fig. 14:  A sample of ICE-CAMERA images  of the 14 classes of ice particles used to train the CNN.

In the following scheme I tried to fit the classes chosen for ICE-CAMERA with the classification scheme of the ice particles

of Kikuchi et al. (2013), an updated version of the original classification of Magono and Lee (1966).

-**Needles:** covering the classes C1a,C1b,C3d              (Kikuchi et al. 2013)

-**Bullets:** covering the C4b-C4c classes.

-**Columns:**  columns covering classes C2a, R2b, C3a, C3b.

-**Hexagonal plates:**  covering classes P1a, P1b, P1c, P4f, G2a, G3a, CP3f, CP3d.

-**Trigonal plates:**  covering the class G2b.

**-Irregular plates:** plate-like particles with irregularities, riming, overgrowing plates, etc. But keeping a basic hexagonal
shape, covering P6a, P6b, P7a, CP6d, R1b, R2b, R2c, R3a, G4b.
**-Rosettes:** bullet-rosettes or column-rosettes, with a minimum of two branches, covering C2c, C3e, C4d
**-Irregular rosettes**: rosettes with irregularities, riming, but preserving the typical stellar outline of rosettes. Covering
classes P7a,P7b,CP2d ,CP4c,CP5a,CP6e,CP6f,CP6g,R1d
**-Irregular grains:** covering CP3e, CP5a, CP6d, G4c,G4a,I3a,I2a,I1a,H1a,H1b
**-Spheroidal:** particles with spheroidal or spherical appearence, covering H1a, H1c. (Large particles with D>600μm)
detected as 'spheroidal' in DC are usually artifacts caused by defocused images and are not considered in the statistical
analysis.
**-Compact columns:** short columns covering classes G1a, C3a
**-Clusters of particles:** covering A1a, A3a, H2a, H1b, P8b, CP3e, CP5a, CP6h
**-Frost**: frost formed on the DS CP7,CP8,CP9
**-Fibers**: non-volatile fibrous material (from local human activities, Styrofoam particles, textile particles, dust, etc)

The last two classes are not considered in the statistical analysis of ICE-CAMERA data: they are just used to detect occasional
frost formed on the DS in case of super-saturation, and man-made, non-evaporable (thus persisting on the DS) materials.
Uncommon ice particle typologies present at Concordia were not considered in the present work. Trigonal plates have been
included, although they are rare, simply because they are seemingly easy to detect with CNN.

### 4.2.2 The training dataset.

For the training of a first CNN, a set of 5500 ICE-CAMERA segmented images of single particles, sampled randomly from
the 2014-2017 ICE-CAMERA database, have been manually sorted into 14 image data stores, corresponding to the 14 classes.
Fourteen of the computer keyboard keys were marked with the symbols of the 14 classes in order to expedite the manual
classification of the initial training dataset. These images were used for a first CNN training. 10% of the images were dedicated
to validation, 10% for testing, and the remaining 80% for training. The first CNN was used for the classification of the ICE-
CAMERA dataset for the years 2014 to 2017. In addition to the classification, the individual crystal images were also sorted
and stored in 14 folders, according to the CNN classification. Selected images from these folders were manually reclassified
into all 14 classes (when misclassified by CNN) and added to a second CNN training dataset. Also misclassified images were
thus re-labeled and used in the training dataset for the new CNN. In this way, a potential positive bias of the confusion matrices
due to the exclusion of misclassified images in the new training dataset was avoided. In the selection of the new training
images, care was taken to ensure a balanced number of training images in the 14 classes. The updated image dataset was
finally divided into validation (10%), test (10%) and training (80%) datasets for training a second CNN. This process was
repeated three times to expand the training database and thus improve the overall precision of the CNN classifier.
Figure 15 shows the final number of training and test images selected for each class. The total number of images used for the
training was 81800. Trigonal plates were rare, and their number in the training dataset was thus artificially augmented by
duplicating the training images, in order to avoid their absence in the small (64-images) training mini-batches.

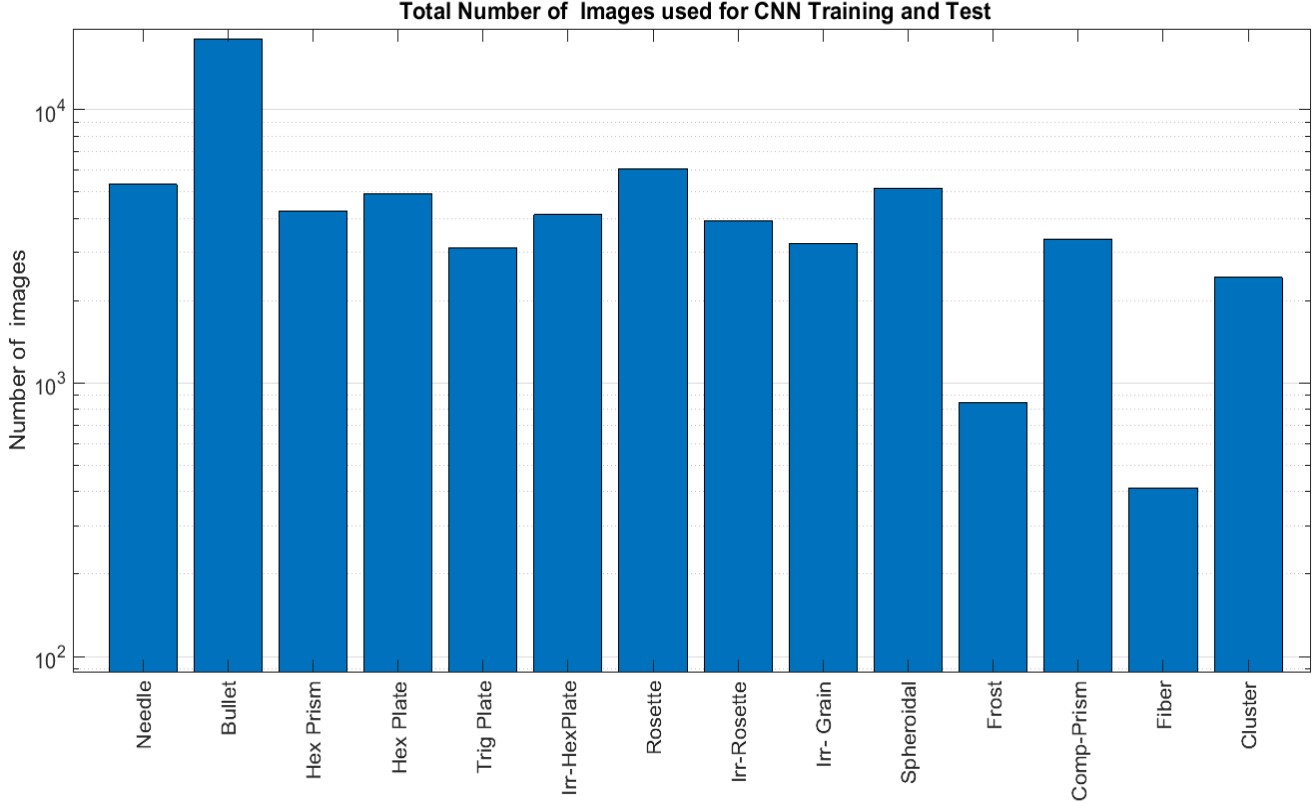

**Fig. 15: The final number of images used for CNN training + validation + test.**

**4.2.3 CNN training details.**
To meet Google's input requirements, all images of single particles were resized to 224*224 pixels. In the training process,
'data augmentation' was applied to the original dataset. Artificially 'augmenting' the image dataset has been shown to be
effective in CNN training (Shorten and Khoshgoftaar, 2019). Images inside each mini-batch are automatically, randomly
'augmented' in order to reduce CNN overfitting. The following transformations were used in augmentation:
- X, Y reflection
- random X, Y translations ±30 pixels
- Random scaling 80-120%
Other changes such as rotation have not been introduced since the ICE-CAMERA images to be classified are
typically oriented horizontally by the image processing procedure (e.g. Fig. 13)
The following learning options were utilized in GoogleNet training:
Solver: stochastic gradient descent with momentum (SGDM)
activation: softmax
Number of Epochs=5
Learn Rate=0.001
Batch Size=64
L2 weight regularization factor=0.005
Validation frequency= every 30 iterations
Shuffle of the dataset at every epoch
The evolution of the CNN training in terms of accuracy and losses is presented in Fig. 16. The validation line closely tracks
the training line, showing the absence of overfitting.

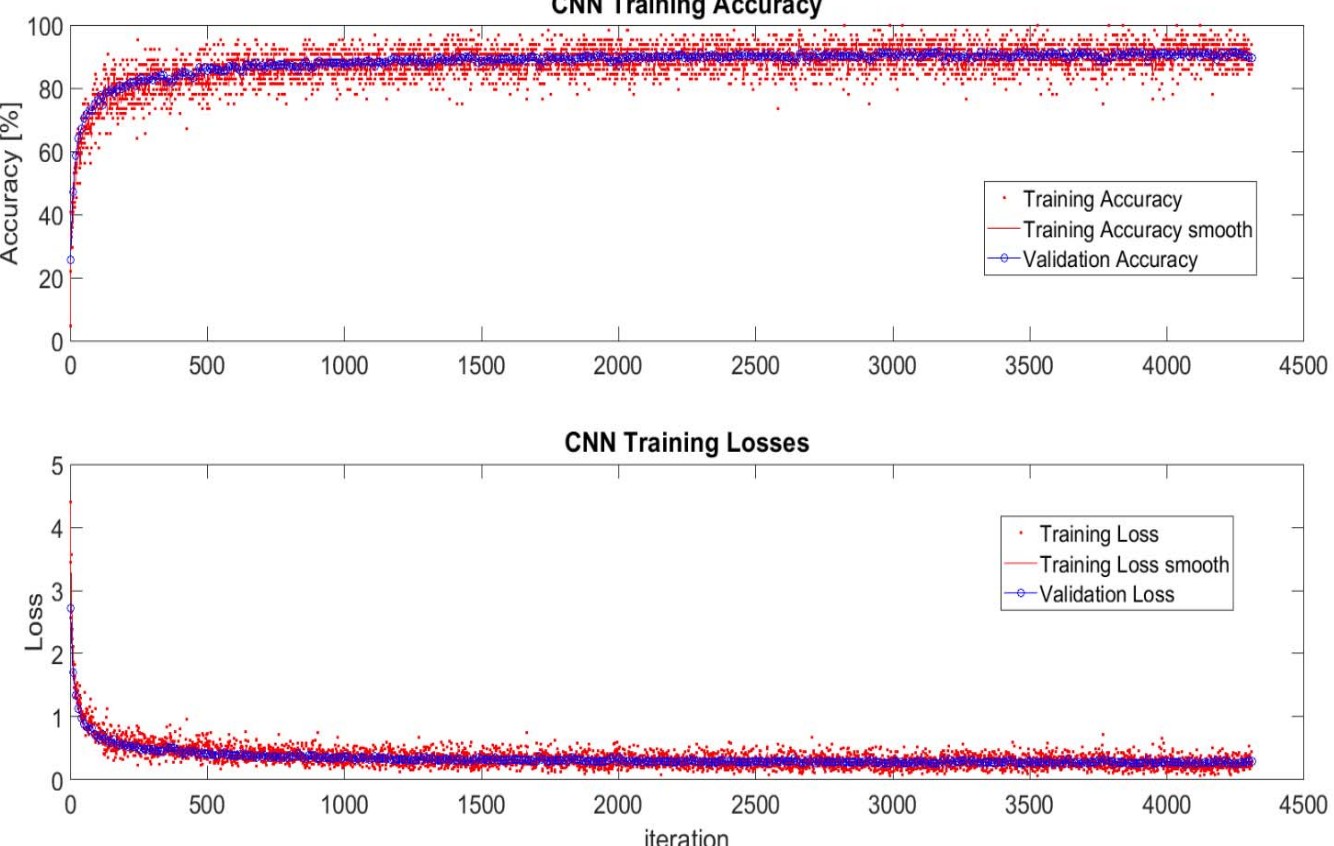


**Fig. 16: Evolution of the CNN training.**

**4.2.4. Testing the CNN classifier.**
CNN's performance test results are summarized in confusing matrix graphs like Fig. 17a.  Each row corresponds to a
predicted class (Output Class) and each column corresponds to a true class (Target Class). Diagonal cells refer to correctly
classified observations. Off-diagonal cells are improperly classified observations (red color markings increasing
misclassification). The column on the far right of the plot shows the percentages of all the examples predicted to belong to
each class that are correctly and incorrectly classified (positive predictive value and false discovery rates, respectively). The
row at the bottom of the plot shows the percentages of all the examples belonging to each class that are correctly and
incorrectly classified (true positive rate and false negative rate, respectively).


    **4.2.5 Accuracy of the classifier**.

In the column-normalized  summary (Fig.  17a), the percentages along the i-th column shows the probability (P) of a "true"
particle in class i-th  being classified in each of the 14 output classes.

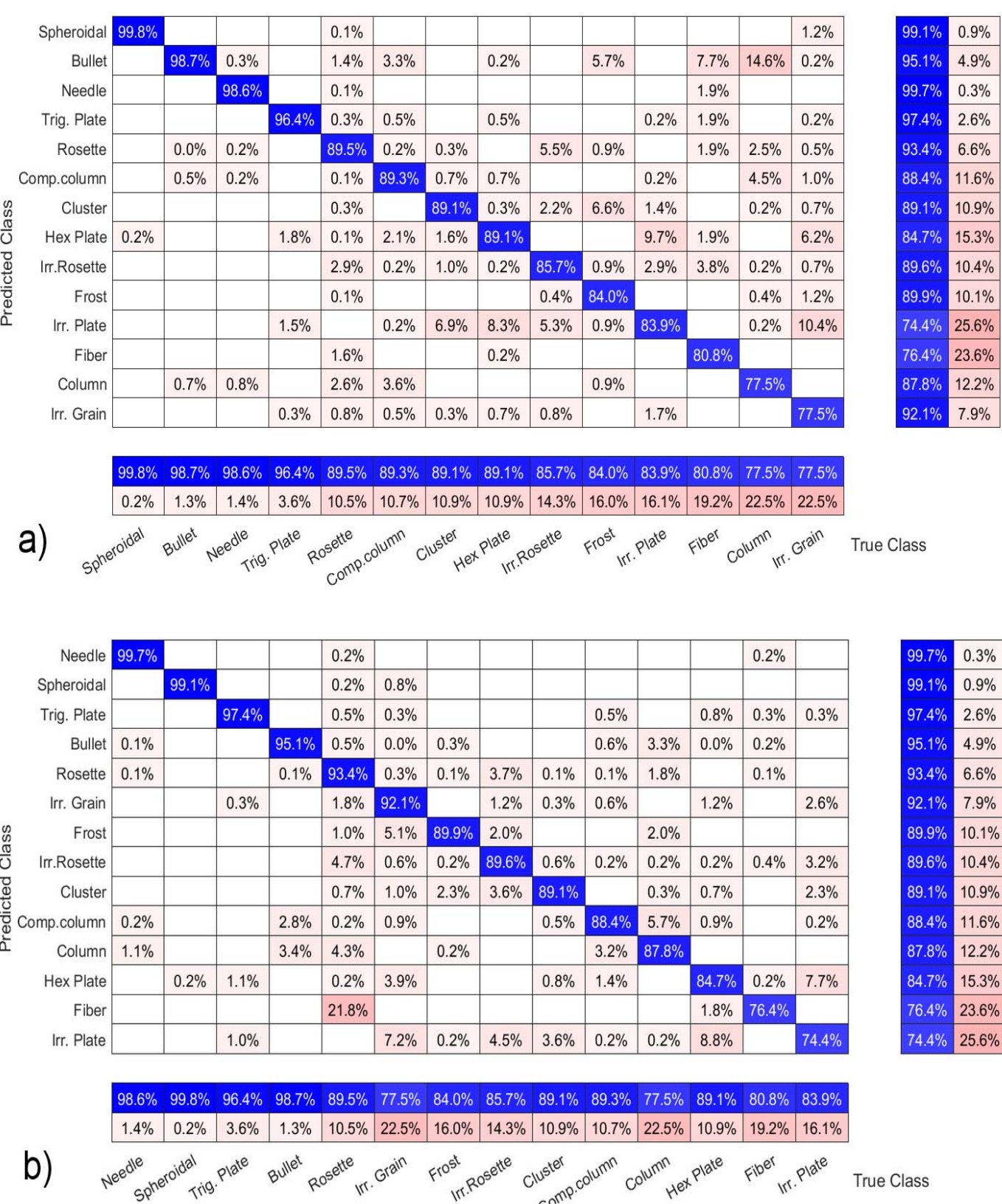


**Fig. 17: Confusion plot of the CNN :** a) **column-normalized, b) row-normalized**

Reading the columns of Fig 17a from left to right, the accuracy of the CNN in properly classifying a particle belonging to the
i-th true class (bottom row) can be assessed. The results are summarized below:

-Good accuracy (P>90%) in identifying needles, spheroidal, bullets, trigonal plates.
-Compact columns are misclassified into columns (3% of the time) and bullets (3% of the time).
-Hexagonal and irregular plates are confused approximately 10% of the time. This is expected since the edges of the plates
(usually small) are sometimes blurred in the image.
-Irregular rosettes are misclassified in 5% of cases as pristine rosettes and in 5% of cases as irregular plates.
-Irregular plates are confused with hexagonal plates 10% of times.
-Irregular grains are sometimes mistaken with irregular plates (10%) and hex plates (6%).
-Columns are misclassified as bullets 15% of the times.
The three-dimensional structure of the ice particles is lost in the ICE-CAMERA images, so that some thick ice forms such as
C4a, P1b, G3b, CP1a, etc. (Kikuchi et al, 2013), if any, are likely to be misclassified by this CNN.
A different view to read the CNN test is the row-normalized summary of the confusion matrix (Fig. 17b).
Percentages along the i-th row now show the probability for a particle classified into the i-th class to effectively belong to
each of the 14 true classes. Reading the rows of Fig. 17b from top to bottom, results are:
-Particles classified as needles, spheroidal, trigonal plates, bullets, pristine rosettes and irregular grains effectively (P>90%)
belong to their class.
-Particles classified as irregular rosettes have a 5% chance of being regular rosettes
-Particles classified as compact columns have a 6% chance of being columns.
-Particles classified as columns have a 4% chance of being a 2-branch rosette and 3% of being bullets or compact columns.
-Particles classified as pristine plates have a 4% chance of being irregular grains.
-Particles classified as irregular plates have a 7% chance of being irregular grains, 9% hex. plates and 5% of being irregular
rosettes
**5. Results.**
**5.1 Overview of  ICE-CAMERA dataset.**
From January 2014 to December 2021, ICE-CAMERA has segmented a total of 11.007.543 particles. This gross count includes
particulates successively rejected for the statistical analysis. Some whole scans were eventually ignored because of poor focus,
sledge motor failures, or the presence of layers of snow or frost. Individual particles were omitted from the analysis due to
their small size or defocus. The distribution of the number of particles observed during the months is shown on Fig. 18a. The
number of scans per month is shown in Fig.18b. Under optimal conditions, one scan per hour is planned, with a typical total
of 740 scans per month.  Some months, problems with ICE-CAMERA, focusing, or processing software resulted in the small
number of scans or particles observed. In most other cases, scans were not recorded when fewer than ten particles were detected
on the DS.

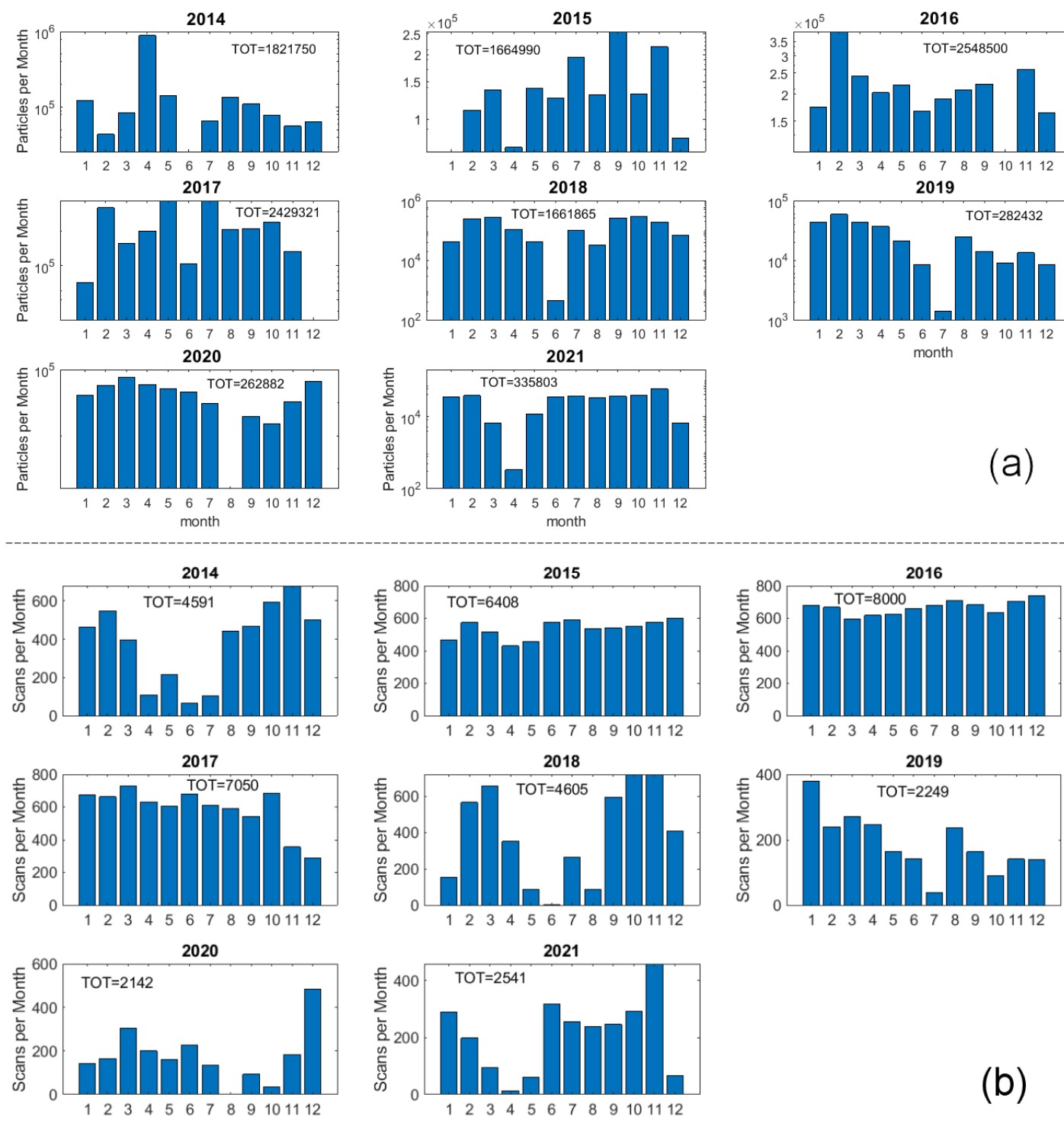

**Fig. 18: Statistics per month for the years 2014 to 2021. a) Ice particle counts per month (total counts per year are also reported). (b) Number of scans per month (total number per year is also reported)**

The number of particles per scan (NpS) is a rough indicator of the intensity of the collected precipitation, but it could be affected by sublimation, because in condition of 'warm' air the smallest particles could disappear from the DS before being detected (sec.3.2). Figure 19a shows the NpS in relation to the air temperature for the whole period 2014-2021, in box and whisker format. On each box, the middle mark indicates the median, and the lower and upper edges indicate the 25th and 75th percentiles, respectively. The lower and upper whiskers indicate an interquartile below the 25th percentile and an interquartile above the 75th percentile.

Most ice particles were detected at temperatures between -60°C and -45°C, characteristic temperatures in spring and autumn.
The NpS at -70°C is not statistically different from the NpS at -30°C. This observation shows that a statistically important
number of particles is measured also at the highest DC temperatures, when sublimation is expected (Sect.3.2) to rapidly deplete
the number of collected ice particles. According with the DC air temperature statistics (fig.19d), most particles were detected
at temperatures above the median DC temperature.
Looking at NpS statistics with relative humidity (Fig. 19b), most particles were detected with relative humidity ranging from
40% to 50%.
Figure 19c shows NpS in relation to wind velocity: ice particles were collected by ICE-CAMERA under all wind conditions
encountered in DC. Ice particles were numerically more abundant when the wind was between 7 ms$^{-1}$ and 15 ms$^{-1}$. As the
average surface wind speed at DC resulted around ≈6 m s$^{-1}$ for the measurement period (Fig.19f), particles were collected on
the DS preferentially with winds stronger than the average, a condition typically encountered in winter in coincidence with
warming events (Argentini et al.,2014). These winds exceed the threshold value of 5 m s$^{-1}$ for blowing snow at ICE-CAMERA
altitude, and may ultimately contain some drifting snow. The drop of NpS for wind speeds above 15 m s$^{-1}$ (very rare in DC)
is probably due to the limited attachment of snow to the DS with strong winds.

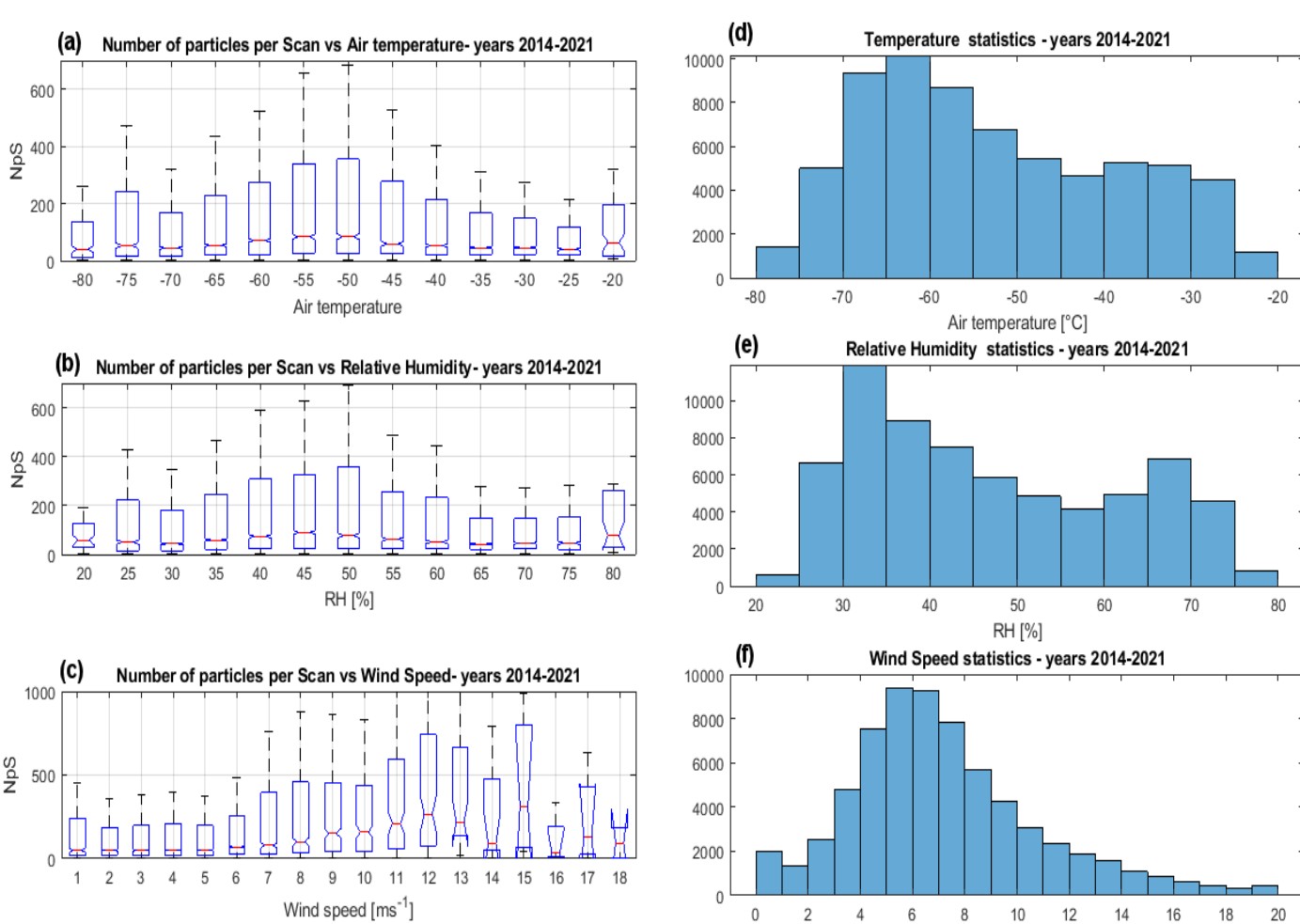


**Fig. 19: NpS statistics in relation to a)Tair, b) RH , c) wind speed.**
**For comparison, the statistics for d)Tair, e) RHy, and f) wind speed are shown for the same period (2014-2021).**


 **5.2 Image processing and CNN used on ICE-CAMERA data.**

MATLAB post-processing software, including the CNN classifier (Sect.4.2) and measurement tools (Sect.4.1) has been
applied to the 2014-2017 ICE-CAMERA dataset. Even if the detailed analysis of these data is the task of a separate paper, a
sample of the capacity of the instrument is presented in this section for the first two years of measurement (2014-2015). The
total particles analyzed resulted in N=553.358. The number of particles classified in the 14 classes is reported in Fig. 20. The
relative rarity of trigonal plates and spheroid particles is evident.

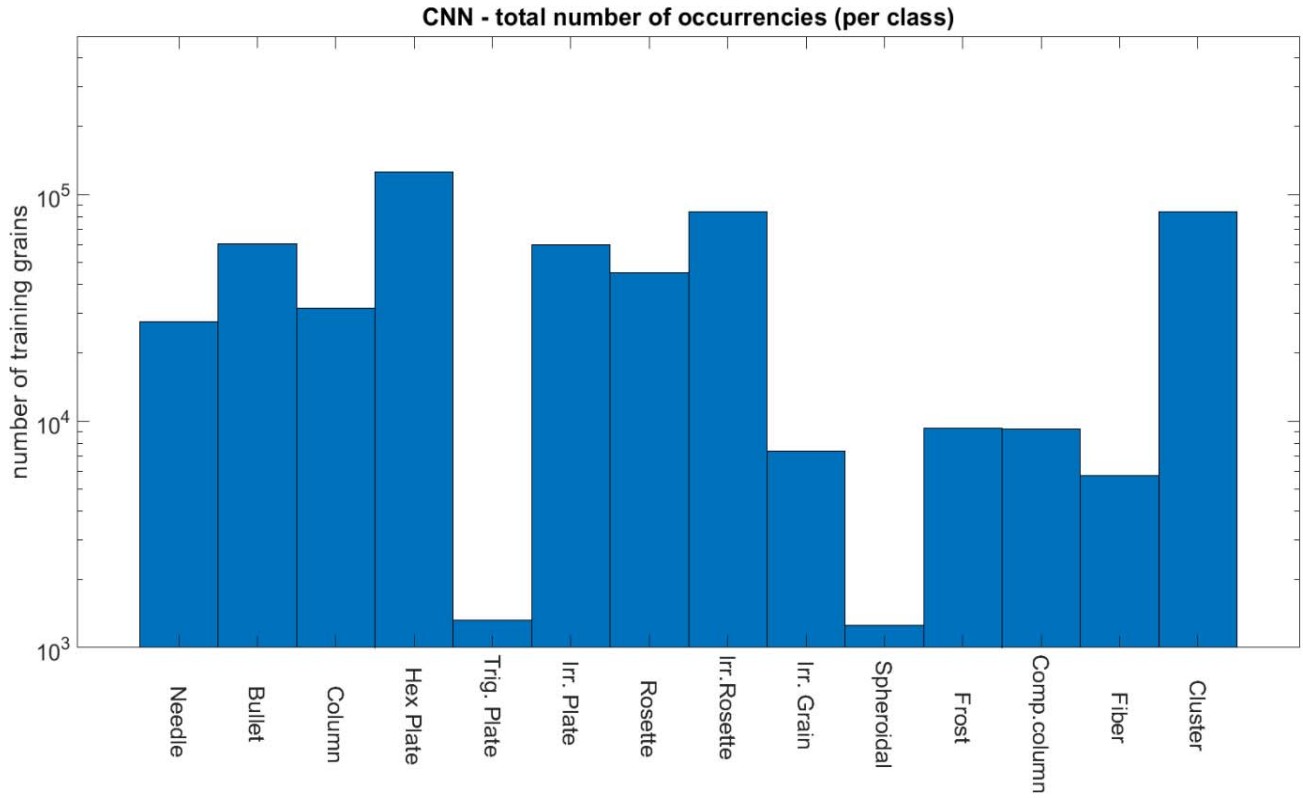


**Fig. 20: Total numbers of particles classified in the 14 classes for years 2014-2015 .**

Figure 21 shows the Feret length statistics in box and whisker format. Particles classified as plates, needles, compact columns,
spheroidal and irregular grains gave an average length lower than 300 μm. Bullets and columns mean length resulted in the
400-500 μm range, while for rosettes and irregular rosettes was in the 350-550 μm range.




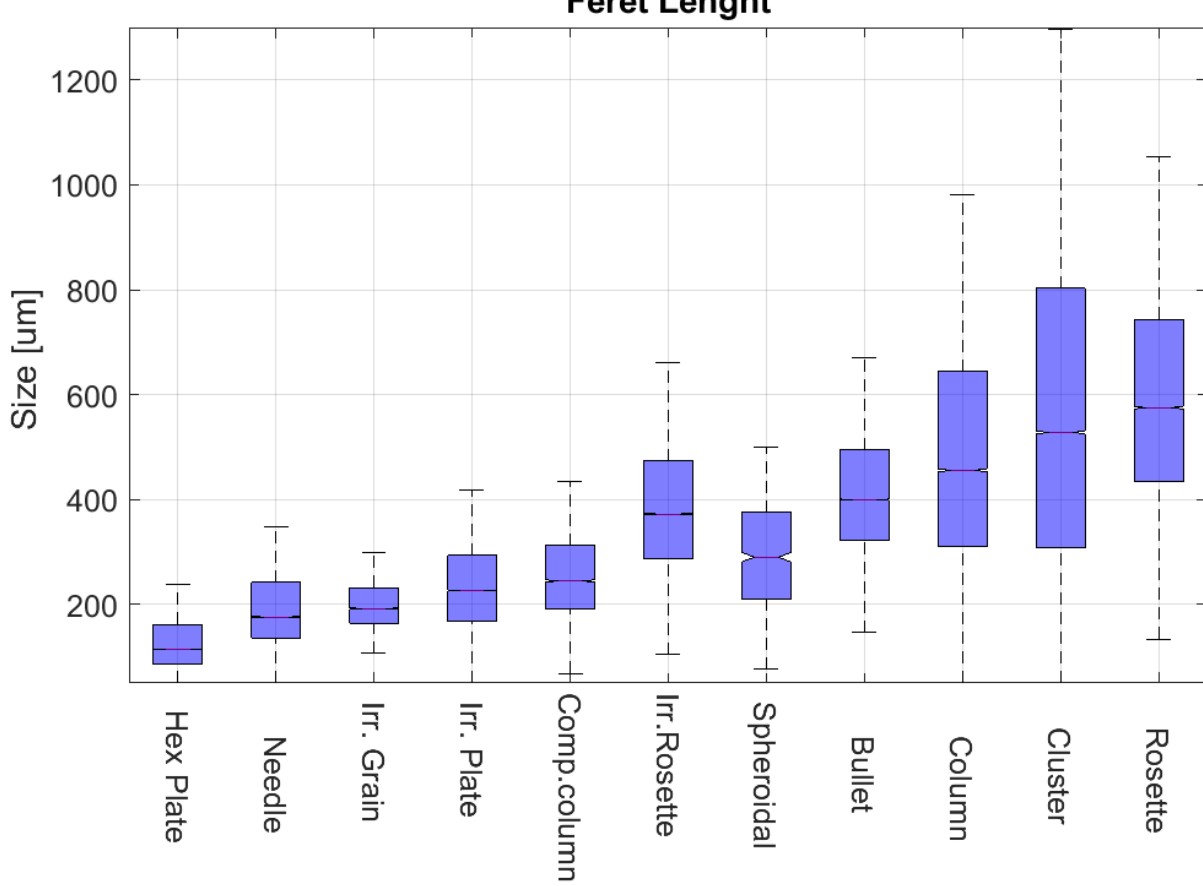

**Fig. 21: Feret length statistics for the years 2014-2015**

Figure 22 shows in detail the probability distribution of the Feret length for plates and rosettes. For plates (Fig22b), the peak of the distribution is for Lferet=100 μm, similar to the peak of the diamond dust (maximum) size distribution measured by Lawson et al. (2006) at SPS in summer (it must be pointed out that Lawson et al. (2006) measured also particles as small as 30 μm , while particles below 60 μm are not processed by the ICE-CAMERA software, and are therefore missing from the probability distribution of Fig22b). This finding suggests that sublimation of particles less than 100-200 μm in diameter during the deposition period (Sect. 3.2) is not relevant for shaping the final particle size statistics. The loss of particles in the lowest size range for sublimation cannot be quantitatively assessed from these data. Nevertheless, the first effects of sublimation are expected to be evident in the ICE-CAMERA images of small particles in the form of loss of sharp edges, eventually leading to spheroidal shapes (sect.3.2). An overview of the images collected in the 2014-2015 summers (where sublimation is most likely to occur) indicates that this effect is rarely observed.  Either sublimation is slower than expected from the simulations of  Sect.3.2, or  what is observed in the summer images of  ICE-CAMERA is just the result of the crystals felt just before the scan,  with the majority of previously fallen small particles definitely sublimated and not detected by image segmentation. This ambiguity will be resolved in DC by taking a continuous serie of acquisitions of  DD in summer conditions. The results obtained from ICE-CAMERA for pristine rosettes (Fig22a), differ considerably from those of Lawson et al (2006), because the peak of the probability distribution resulted L=480 μm, to be compared with L=120 μm of Lawson et al (2006). This difference is not explicable with the eventual sublimation of the smallest rosettes on the DS. Instead, this result is a realistic feature, sustained by the direct visual observation of rosettes in DC precipitation. A much greater amount of rosettes is actually observed in DC during precipitation from clouds than during diamond dust events. Even if rosettes in diamond dust are much

smaller than rosettes from clouds, the numerical dominance of cloud rosettes explains the large median value of their Feret
length.

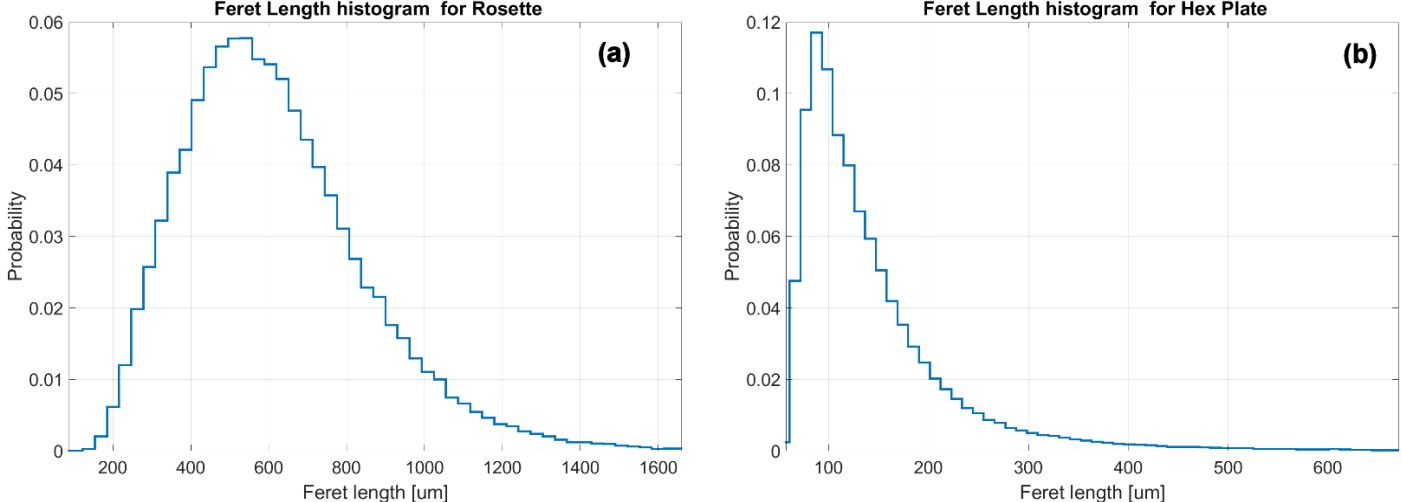


**Fig. 22: Feret length probability distribution for a) rosettes and b) hex. plates. The relevant presence of small plates**
**(D<200µm) suggests that sublimation on the DS is not relevant. years 2014-2015**
Figure 23a shows Feret's aspect ratio per class. Not surprisingly, many "rounded" classes (plates, rosettes, etc.) have an AR<2.
Compact columns show a median AR close to 2.4, while columns and bullets are close to 3. The average AR for the needles
was 3.2, which is lower than expected for the reasons outlined in section 4.1.3.

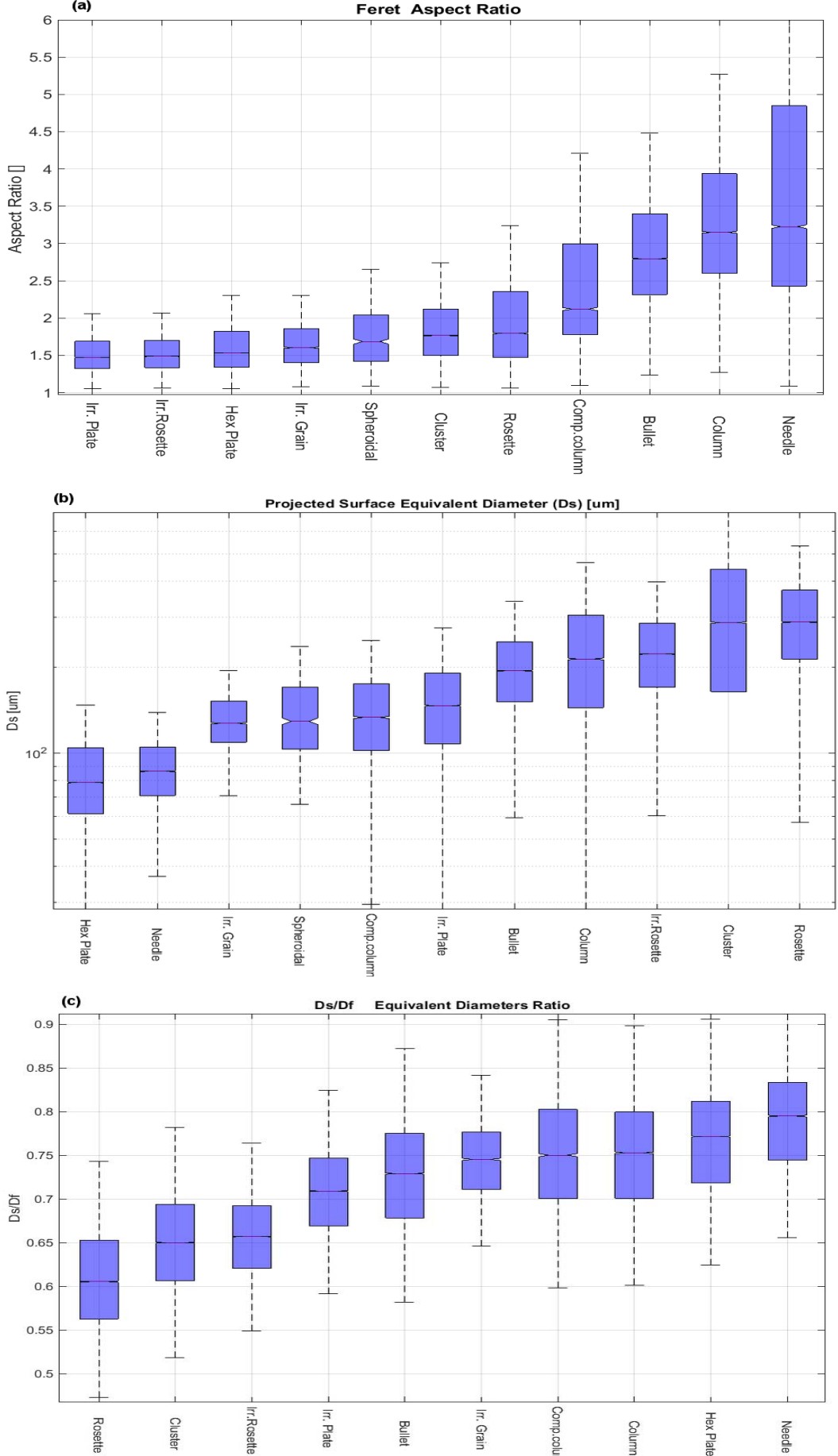

Fig. 23: Statistics of (a) Aspect Ratio, (b) Projected surface-equivalent diameter (Ds), (c) ratio between surface-equivalent (Ds) and Feret box-equivalent (Df) diameters for the years 2014-2015.


Figure 23b shows the surface-equivalent diameter Ds of the particles. Figure 23c shows the ratio between the surface equivalent diameter (Ds) and the Feret-box equivalent diameter (Df). The difference between the two diameters is relevant for "fluffy" particles like rosettes and clusters. For those particles, Ds/Df gave values of 0.6 to 0.65. For comparison, a round particle is expected to have a ratio of Ds/Df=0.89.

## 6. Conclusions.

ICE-CAMERA, although very similar to a simple flatbed scanner in its basic design, has represented a technical challenge for its implementation at DC. Hardware and software have been continuously and extensively modified at DC over the past five summer campaigns. The result is now a reliable instrument, running throughout the year on an hourly basis, for the statistical study of precipitation in internal polar areas. Particle size and morphology are automatically obtained, and some semi-quantitative precipitation estimates can be derived. The collected data are automatically pre-analyzed, but they can be post-processed at any time, in order to follow the continuous improvements of the image processing and machine learning algorithms. The GoogleNet CNN, trained specifically for this instrument, has succeeded in classifying ICE-CAMERA images into 14 form classes, with an accuracy of more than 80% for most of them. The instrument is particularly useful for automatically measuring the size of individual ice particles in precipitation, a process virtually impossible manually, and certainly impossible on the field in DC and elsewhere on the Antarctic plateau in winter. ICE-CAMERA scans are carried out every hour. Keeping the surface of the instrument free of frost all the time and cleaning it by heating the deposition surface after each scan is paid with the possible loss of small ice particles. Particles less than 100-200 um can disappear by sublimation before being recorded, especially in summer. This problem is complementary to the problem encountered when observing precipitation manually: when observing precipitation manually every 24 hours, (as is the case of DC) the reprocessing of particles, or the formation of ice and hoar artifacts cannot be prevented. In ICE-CAMERA, frost and ice regrowth are suppressed, but small particles may disappear for sublimation. The effect of sublimation on particles observed with ICE-CAMERA cannot be easily quantified, also given the broad range of atmospheric conditions encountered by the DS throughout the year. Images of small particles (100-200 μm) (such as plates) collected during the 2014-2015 summers rarely show evidence of early sublimation such as edge smoothing or rounding. While encouraging specific experiments with ICE-CAMERA, this observation suggests that sublimation could be slower than predicted by simulations. ICE-CAMERA data, collected since 2014, have already been statistically processed and the results will be described in a specialized paper. Results from a subset of data (years from 2014 to 2015), was presented in this work. As preliminary results, in DC the rosettes were found to be significantly larger (480 μm) than those observed at SPS by Lawson et al (2006), while the plates were of similar size (120 μm). These results demonstrated the capability of the instrument to classify and size individual ice particles in DC precipitation. Unfortunately, only non-polluted, very cold, low humidity, low precipitation environments (like high mountain tops, dry polar environments) could house a similar instrument. In the presence of pollution, marine aerosols or dust, manual cleaning of the DS would be required to remove solid particles and salts escaping sublimation. For coastal zones, the temperature is generally close to zero, making the thermal cleaning of the DS by sublimation problematic. In these environments, if an instrument like ICE-CAMERA were installed, a mechanical wiper would replace the heated glass of the current instrument. Furthermore, the CNN presented in this paper should be re-trained with different classes of ice crystals.





**7. Technical issues.**

Using ICE-CAMERA at DC, as well as other automated instruments, was difficult. The instrument had several failures along years, and each one was difficult to fix, at least in winter, when the instrument had to be dismounted from the roof of the shelter at -70°C and eventually fixed in the local lab by the winter-over crew, with remote assistance from Europe. Until a few years ago, communicating with DC was limited to email with small attachments, making remote assistance a lengthy task. Even today, connecting the rest of the world remotely with the ICE-CAMERA PC, to operate with the instrument software, is virtually impossible. Most hardware failures in DC were due to software bugs or computer failures. Rather than having trouble with low temperatures, operating in DC means dealing with limited heat-dissipation of PC parts such as power supply and hard disks, electrostatic discharge issues in low-humidity, heated environments, lack of spare parts for most of the year, a varied skill-ness of winter-over personnel. Failures in the thermal control of ICE-CAMERA caused some mechanical stress and failures in the focusing sledge, while water condensation eventually rusted the bearings of the stepper motors (all bearing were de-greased for a better low temperature operation). The CNN used to classify ICE-CAMERA images is continually changing and improving and the CNN training dataset increases with time, as new images collected by ICE-CAMERA are used as new training ones.

**8. Code and Data availability.**

The CNN developed as part of this work (under Mathworks MATLAB R2020B), along with the image dataset (224*224 images for the 14 classes of particles) used for training, validation and testing the CNN are available in the ZENODO repository (Del Guasta, 2022)

**Acknowledgements.**

I am grateful to the Italian Antarctic Project PNRA for supporting this work with the projects ICE-CAMERA (PNRA 2009/A4.1) and PRE-REC (PNRA 2013/AC3.05). I am also grateful to the 'Osservatorio Meteo-Climatologico Antartico' (PNRA 14_00100) for the Meteo data, and to all the logistics staff and winter-over crews of Concordia station, all working hard to permit our scientific activity. I am also grateful to Francesco Castagnoli (INO CNR) for the initial design of the instrument.

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
