# Peer review of "ICE-CAMERA: a flatbed scanner to study inland Antarctic polar precipitation."

_Atmospheric Measurement Techniques, 2022_

## Author Comment (AC2)

1)Abstract is not adequate.

A new extended abstract is under preparation

2) L77-78: "The principle is simple: at low temperatures and low wind speeds (conditions encountered at DC), precipitation falling over a flat glass accumulates with time and remains frozen until it sublimates, …"

corrected

Language should be improved, but also the collection and issues with collection efficiency should be discussed here or later in the paper. Particles deposit on the DS, but what makes them stay there ("remains frozen")? What are the effects of wind both during collection (deposition) and after collection (can particles be removed by wind)? Could any biases be introduced based on wind effects or other effects depending on size or shape?

A discussion about 'what makes them stay there ' was added. About the effect of wind on particle deposition and removal I instead admit that this cannot be modeled properly using my (and literature) knowledges and tools, also because the wind effect is complicated by electrostatic effects occurring on the dielectric DS of ICE-CAMERA, the effective wind speed close to the DS surface is not known, (no micrometeorological data are available there). This would be the topic of a more specific work. I just added a discussion about the wind-speed threshold for wind drift (and thus for the removal of particles from the DS…) based on specific Dome-C results of Lebois et al., 2014, Etienne et al.,2016 .

3)1.2 Camera/1.3 Focusing and 2.1.2 Limitations:

Say something about resolution (resolving power). Pixel size corresponds to 7 um, but resolution does not seem to allow detecting particles smaller than 70um (70um, not 60um as you state in L333, is equivalent to 40000um2).

Corrected:

How much does the focus change between calibrations?

How long does focus calibration take? (Could it be done before each scan instead of 6h intervals?)

Corrected: The typical focus adjustment between two calibration if between 0 and ±0.25 mm. The calibration takes approximately 10 minutes and for this reason it is not performed after each measurement in order to spare PC resources for data processing.

Consider including a discussion of accuracy of provided particle sizes.

A discussion about the calibration of the measurement on the DS was added

4)Thermal control:

 L168-170: "The NI-Labview software controls the internal temperature of ICE-CAMERA above -40°C, and the DS temperature always under -5°C ("Heater 1" in Fig.3)."

Is this always possible?

Specify if this is general heating of ICE CAMERA or specific for sublimation (then refer to Sect 1.6.1).

**Corrected:** **These  conditions are maintained all-year round in all phases of the measurement cycle**

Unclear where Heater 1 and 2 are.

Heater 1 and 2 swapped in Fig 3.

Corrected

L179-181: "An indoor test (Fig.7), showed a heating of rate of 2.5°C min-1, and a cooling rate of 1 °C min-1. The cooling rate is almost 50% of the heating rate just due to the sandwich heating-glass structure, with the heating layer at the middle"

"…just due to the sandwich...": is this passive cooling, what is "heating layer at the middle"?

Corrected: I also introduced a description of the 'sandwich' and its thermal characteristics

Is the indoor test really relevant here? You then report the heating/cooling rates outside. So, Fig7 could maybe be replaced with a cycle outside.

I prefer to keep this picture, as in steady-air the same thermal cycle (with just a different offset temperature) occurs outside, as the thermal constants of the instrument are always the same.

In Sect 1.5 you mentioned about heated air between glass sheets (DS and second sheet) and keeping DS frost free; is that the actual heating of the DS?

Corrected: introduced a description of the 'sandwich' ant its thermal characteristics

Detail: how much air pumped inside and through the space between the windows?

Done: The outside air is pumped (through a 3.5 l/m miniature pump) for  five minutes every hour inside the instrument

Distance between glass sheets?

Corrected: 13 mm

When air between glass sheets is not pumped, is it sealed or can air circulate?

Done: sealed

L192: "Heating is anyway interrupted if the DS temperature exceeds -5°C…"

How often does this happen?

Only in warmest summer days

5)Deposition/scan/sublimation cycle:

The cycle of deposition, scan, and sublimation is not properly explained in detail. The reader gets the details somewhere between the lines.

Corrected: better described the cycle in the 'overview'

L190-192: "…sublimation of the majority of particles (D<1000 um) is complete within 20 minutes, with just a few big (D>>1 000 um) grains still present after 30 minutes.

After these tests, the heating time was set at 10 min."

"D" is not defined.

Corrected

"within 20 minutes": 20min after the 10 min heating period (after 10min heating was turned off)?

Corrected

"few big grains" Can you be more specific? After 30 min grains were larger than 1mm? Or ice particles that original had D larger than about XY mm?

Corrected: (initial diameter>>1000 um)

Then you continue to discuss the cycle and for how long the DS is "sensitive to falling ice particles" and that this time is variable depending on conditions and that there is an uncertainty associated with this.

What is in L195 the "evaporative removal of particles during the accumulation period"? Is this wind related; why evaporative now and not related to sublimation; what is accumulation period".

corrected

All these details should be explained more clearly and then the issues discussed clearly too. In particular, make it clear that the deposition phase is not clearly defined (only its end due to the scan). I think I have not seen how long a scan takes. Would scanning more often be a way to study sublimation issues related to smaller particles (100 um)? Would could then see how they sublimate in consecutive scans.

Corrected:
evaporative→sublimation

accumulation period→ 'Deposition period' (now defined)
Scan duration-> 2 minutes

-Would scanning more often be a way to study sublimation issues related to smaller particles (100 um)? Would could then see how they sublimate in consecutive scans.

This is possible, and can be planned (in summer), as a test ,on a few target particles, but in this specific instrument it cannot be done (and was not done) routinely, because the image processing of a typical population of one hundred ice particles takes 10 minutes or more. Small DD particles such as small (<100um) plates, the most interesting in a sublimation test, are very scarce in summer Moreover, in winter the instrument works unattended, and experiments of this type are virtually impossible.

6)Fig 8:

What measurements does the figure show? For one 1h cycle, how many temperature measurements are shown?

Fig 8 left) at what wind speed?

+25degC at still air, +20degC at 8m/s: are those temperatures averages, median, or ? (I can see a large range, maybe 20degC to 35degC.)

Wide spread in blue; two regions (sublim and deposition phases)?

Left: better also show dT instead of DS temperature.

Could Fig8 be separated in one fig. only the deposition periods of 20 min prior to scan, one for after/during sublimation heating? This could be clearer instead of heaving heated and not heated temperatures mixed in one figure.

The figure was in fact confusing, and was totally removed

7)Sect 1.6.2 Sublimation of ice particles

You discuss sublimation, both unwanted and wanted sublimation here. When saying "negative effect" you refer to the unwanted sublimation. Be careful to keep clarity in this section.

Corrected

L214: "as DS ice is always super-saturated relative to the surrounding air"

Wrong as it is stated. It is also unclear what "DS ice" refers to.

At the heated DS the equil. vap. pressure will be higher; thus the surrounding air will have a lower RH at the heated DS and ice on the DS will experience sub-saturated conditions.

L215-216: "the vapour pressure of ice on the DS relative to the surrounding air (saturated relative to ice…"

Wrongly or unclearly stated: "the vapour pressure of ice on the DS..." should be something like:

"the vapour pressure saturated relative to ice at the DS temperature".

Fig 9: Why show this ratio (e0ice(T_DS)/e0ice(T_amb))?

More intersting would be the inverse (e0ice(T_amb)/e0ice(T_DS)), i.e. RHice at DS if surrounding air is at RHice=100%. For dT=5K this would range between 45% at -80degC to 65% at -10degC. It shows the sub-saturation. Fig 9 is not further discussed or used later on. If you want to keep it, I suggest to plot inverse.

L214…..Fig.9 and its discussion were eliminated, as unnecessary: the simulations of sublimation contain  the same information and lead to more readable results.

You use over saturation, oversaturation, over-saturation, and super-saturation. Only use one term for clarity, I would suggest super-saturation.

Corrected adopting super-saturation (thanks!)

L228: "growth rate of the facial area" Growth perpendicular to facets?

Corrected

L229-230: Unclear what critical super-saturation and critical temperature are.

Sentence removed

L231 "0.05% to 5% under-saturation". How is "under-saturation" defined here?

Corrected in 'sub-saturation'

L234 "The steady-state shape of the sublimating crystal depends"

Unless explained or defined, it seems contradicting: a sublimating particle is not in steady state.

Text was modified and 'steady-state shape' removed

L246-247 "The simulations assume the completion of preliminary sublimation of points and edges of the particle"

Unclear language (preliminary, points, edges).

Changed: The simulations assume that the preliminary sublimation of  the high-curvature parts of the particle was already completed

L252 "still evaporate" Unclear what the "still" refers to => "sublimate".

Corrected

L253 "could survive along the heating period" => "will survive longer than the heating period"?

==corrected==

L259: Meaning of "After the sublimation period, DS is exposed to falling crystals."?

DS is always exposed. Particles depositing during the sublimation period may partially sublimate and then be included in "collected" particles. (see comments for Deposition/scan/sublimation cycle above)

==Corrected:During the deposition period, the collected particles also undergo sublimation, although this is much slower than during the sublimation period.==

Fig11: Do you have any experimental data on sublimation times for actual particles (similar to your spheroids)? That would help to set your statement in L270 in relation, or that you only see few particles with partial sublimation.

==This experiment is possible, and can be planned (in summer) on a few particles, but in this specific instrument cannot be done routinely. In winter, the instrument works unattended and experiments of this type are almost impossible. Moreover, small DD particles such as small (<100um) plates are very scarce in summer.==

==Added:== Some small plates (observed mainly in winter, when sublimation in the deposition period is very slow) showed smoothed corners, but it is not clear if this is induced by local sublimation or is a natural shape

8)Sect. 2.1.2

1) Specify better: 2000 per scan?
==yes, corrected==

2) Explain what you mean with "By default". Relate to resolution (see 3) above).

==Corrected==

3) Specify the segmentation error (particles are overlapping?). How can a particle be counted twice (double counting)? What is 12% in your worst case (overlapping particles account for 12% of particles in one scan?)?

==Corrected : 12% of particles in one scan. More details were added==

4) Two issues are brought up together but are rather two limitations that should be listed/discussed separately. It is unclear how and on what basis these spheroidal particles are disregarded.

==Corrected==

9)Sect 2.2.2 Training dataset

I would like to see a better description of the various datasets.

What is the "image dataset" (L437)? How many images; from what time period?

Changed: period 2014-2017. About the size of the dataset, it was already stated in the text, and their distribution among classes was already shown in fig.15

How have these images been selected out of all available images? Randomly or by some other selection criteria?

Changed

This image dataset is then apparently split in 10% validation dataset and 90% training dataset (25705 images).

What about a testing dataset?

Say how testing (see Sect 3) was performed, in Sect 2.2.2 or Sect 3 for example. How many and which particles were used to test the CNN after training is complete, i.e. to produce the results shown in Sect 3?

You are perfectly right: for the testing I used a testing data set including approximately 50% of training images and 50% of unused images. This 'error' was corrected in the new version: now 80% of the whole dataset used for the CNN training was used as training data set, 10% for validation data set, 10% for testing data set. Moreover, I increased the number of training particles for most classes. Trigonal plates are rare, and their number in the training dataset was thus artificially augmented by duplicating the training images for this class, in order to avoid their absence inside most of the small (64-images) training mini-batches. Image augmentation during training contributes to a partial spatial randomization of this special class.

Provide more details about the augmentation: To what size is the original training dataset of 25705 images increased after augmentation? Each class augmented the same (number of images increased by same factor)?

There is no formal answer to this question: Image augmentation is introduced randomly (within the fixed limits and transformation types) to each image of each mini-batch used during the CNN training. Each mini-batch represents a random sample of the data base.

10)Biases during training

 The number in each class is different. This can generate biases in the CNN, please comment. In particular, I think that part of the low performance for trig plates is due to the fact that this is the smallest class in the training dataset.

I increased the number of training particles for most classes in order to better equalize their numbers. Trigonal plates are rare, and their number in the training dataset was thus artificially augmented by duplicating the training images for this class, in order to avoid their absence inside the small (64-images) training mini-batches. Image augmentation during training contributes to a partial spatial randomization of this special class. The resulting CNN is more performant, with 90%

classification accuracy of the training data set just after 3 epochs, compared with 83% (after 5 epochs) of the first version of the paper. The accuracy in the detection of trigonal plates is now reasonable 86%  (on training data set). Sections 2.2.3, and 3 were modified, accordingly.

11)Data

A few example images in the full resolution should be included in the paper. This will help when discussing resolution and sizing accuracy (see Major point 3) above).

Fig.14 already shows examples of good quality ICE-CAMERA images

Reconsider if you interpreted government policies with respect to sharing data correctly. It would be useful to share the whole image dataset. Should that not be possible, then please share at least a sample dataset.

I will share the CNN and sample data-set for each of the 14 class. I cannot share the entire data set as it is still under analysis, and is going to be used for a second paper.  The actual paper is intended to describe the instrument and its software.

Minor comments

==================

Consistency with terminology and spelling in various places.

E.g.

GoogLeNet, Dataset

Corrected both

I would expect that Introduction is Sect 1.

Sect 2 is on instrument; 2.1 Overview (instead of 1.1)....

Corrected

L69: "In this work, the term 'precipitation' will include both diamond dust and"

Should perhaps refer to "precipitating diamond dust".

Corrected

"cooling speed" => "cooling rate"

Corrected

"eventual" => "occasional" (2 x)

Corrected

"deposition window" => "DS"

Corrected

"By the way" is not good English for an article.

corrected

Fig 12: "Reg.growed" Use correct label and be consistent with text.

corrected

Fig 13: Remove title "ICE-CAMERA: Summary of detected grains…"

corrected

Be consistent: what is synthesis image?

summary-image,synthesis image,mosaic, summary image

corrected

L325 "measures" is not a noun.

corrected

L325 and L83: specify what weather data are.

Added a sentence

L384 "hexagonal prism"

I would call this class "long column" or something else with "column"

Fig 14 "compact prism" should be "compact column" (as in L395).

I will follow this indication in the next version of the paper, but not done yet because I must change accordingly most of the labels in the software

L401: Why refer to "insoluble"?

corrected

L451 "3.1 Precision of the classifier" should be "3.1 Accuracy of the classifier"

corrected

L460 "mistaken" = "mistaken with each other"

corrected

L 462 "Compact columns are misclassified almost 20%"

I read 15.4% in the bottom row, not 20%.

A new discussion about the (updated)  CNN results  will be rewritten

L473-474 "The three-dimensional structure of the ice particles is lost in the ICE-CAMERA images, so that some thick ice forms such as C4a, P1b, G3b, CP1a, etc. (Kikuchi et al, 2013), if any, are likely to be misclassified"

Be a bit more specific (or give at least one example).

Even if is quite complicated to find 'clear' examples, I'll try to introduce a specific figure for this problem.

L498-499

"The classifier was used…" You are showing results of the classifier but then also of the "image measurements" for the resulting classes.

corrected

Is the Jan-Feb 2017 period not part of the image dataset used for training/validation and testing?

The image data set used for training/validation includes images from 2014 to 2017 (so, also jan-feb 2017). But just a random sample of the whole image data set was used for training/testing, and only a few images from the Jan-Feb 2017 period were thus used in training/validation

L506 "maximum whisker length is here equal to the interquartile" seems wrong.

Total box height is equal to the interquartile range. What do the whisker lengths indicate?

Corrected, my mistake! Lower and upper whiskers mark one interquartile below the 25th percentile and one interquartile above the 75th percentile, respectively

L507 "some relevant differences" seems to refer to one relevant difference.

corrected

Fig 20: can you include a grid line for 100um?

Yes: To be corrected in the final version of the figure

L532 "diameter of the circle equivalent to the bounding box"

definition is ambiguous:

"diameter of the circle with the same area as the bounding box

Similarly, in L533: ambiguous definition.

corrected

Fig 23: Suggestion: put (as whiskers only) on scatter plot Feret-box vs projected-surface equiv. diameters. In that way direct visual comparison is possible.

L543-544 "Commercial or customized instruments do not have this flexibility, more typical of old-style handcrafted products"

Purpose of this comparison, what instruments are you comparing to?

removed

L548 "convolutive" => "convolutional"

corrected

L552 "precisely" Do you mean precise here, or accurate?

Anyway, neither of the two was discussed.

removed

See Major point 3) above.

Concluding remarks on potential improvements (in particular with respect to the sublimation problem). Use of ICE-CAMERA in other environments (not Antarctica)?

To be added in the final version. Unfortunately, only unpolluted ,very cold, low humidity, low precipitation environments could host a similar instrument (high-mountain tops, dry arctic environments)

---

## Author Response (AR1)

Here are, in yellow, my answers to the Referees' and the Editor' comments. Having changed some text and some figures, the page and figure numbers commented by the Referee are now different, please see the marked text submitted: the major changes are there marked in yellow

 **(Editor)**

 I have done a first brief review and I am convinced that your manuscript contains relevant and interesting information for the community. I have however some suggestions to improve your manuscript before sending it out for review.

1. The Introduction is too short, and should better present the general context of this work/instrument, as well as previous work on similar topics, in order to better motivate the main objectives of this paper. For instance, Schlosser et al, 2017 (The Cryosphere) mentions precipitation measurements at Concordia. Grazioli et al, 2017 (The Cryosphere) present a field campaign conducted at Dumont d'Urville with various instruments dedicated to precipitation, among which a snowflake imager (MASC). The Hydrant observatory (Gorodestkaya et al., 2015, The Cryosphere) has a component related to precip, inland Antarctica. AWARE is also an important campaign in the Antarctic context (Lubin et al., 2020, BAMS).

I improved the introduction partly following the references suggested, and partially adding other relevant references. In fact, my work focus on the photographic observation and sizing of ice particles, a topic rarely frequented on the plateau.

2. The section "Image processing" remains too general and should provide more detailed information, for instance about the different steps and the associated parameters (how many, how their values were selected etc).

I added some details to this part, but basically it was already explained in its basic lines. In fact the image-processing uses 'standard' operators, shape parameters, etc. (I added some references for their definitions) and can be calculated the same way by anybody under any software environment. The number of pixels used in some operations, the sequence of operators udes, etc., are quite a personal choice.

3. The presentation of the CNN requires also more information: number of layers, neurons, activation, loss function etc are not mentioned, so a reader cannot reconstruct the proposed algorithm. The fact that the trained CNN does not over-fit is stated but not quantitatively supported (ex: learning curves).

I really welcomed this request by re-training the CNN  and tuning the training parametrs.. I adjusted a few parameters (e.g. L2 regularization) and at present the CNN is slightly improved with respect to the first submission. Thank you! Confusion matrices are slightly different  now. The learning curve is now shown.

4, I would suggest to add a few basic statistics about the collected data set along the years (occurrence of the different types of crystals, maybe per season) to illustrate the potential of this instrument and the data set.

As the entire statistics is a topic for a second paper (I used ICECAMERA data together with LIDAR and microwave radiometer data for a robust analysis of precipitation data. That's why I split my work into two papers: this one about the ICECAMERA instrument, the other on the statistical data analysis. Anyway, I used  a  subset of ICECAMERA data (2014-2017) in this paper, showing some relevant results and mainly in order to discuss the possible particle

sublimation of small particles

I will be happy to evaluate an updated version of this manuscript.

I spontaneously added a section about my simulation of the sublimation of ice particles, as it occur at the surface of ICE-CAMERA. I hope this will clarify the limits of this technique, that were quite unclear in the previous version

%%%%%%%%%%%%%%%%%%%%%%%%%%%%%%%%%%%%%%%%%%%%%%%%%%%%%%%%%%%%%%%%%%%%%%%%%
%%%%%%%%%%%%%%%%%%%%%%%%%%%%%%%%%%%%%%%%%%%%%%%%%%%%%%%%%%%%%%%%%%%%%%%%%

REFEREE 1:

Major remarks

1. As mentioned, the topic fits well the scope of AMT. For this particular journal a higher level of detail on the instrument itself (see also minor remarks below) is needed. I believe that the author has both data or simulation tools available to better characterize the sampling uncertainties associated to ICE-CAMERA. I would like for example to see statistics on the number of particles collected according to the wind speed (I see data are available in Fig. 8) or according to humidity/dew point which influences sublimation. The author provides interesting hints about sublimation time for individual particles in section1.6.2. The author may apply similar techniques over a populationof particles, for example assuming a PSD (particle size distribution) and repeat the simulations multiple times for multiple PSDs and other parameters temperature for example). A first approximation could be to use a constant PSD/snowfall intensity over the measurement period or to elaborate more realistic scenarios.This is only an idea, and the evaluation could certainly be done in a different way but the objective is the following: by extending the simulation to a population of particles, something more could be said in terms of (theoretical) catching efficiency of the instrument, a key aspect for the readership of this journal.

I particularly welcomed this comment by adding new simulations for an initially uniform PSD of sublimating spheroidal particles. I also added some discussion about the attachment and wind-removal of particles on the instrument window. Some new pictures, based on the ICE-CAMERA dataset , were added for a discussion about effects of wind and temperature on the collection of particles.

2. It was not clear if the main focus of the paper is the instrument or the CNN-based classification. I recommend to clarify it. In either case the level of detail should be increased (see my other comments). My assumption was that the main target is the instrument so I did not comment in much detail about the CNN. In fact, the image-processing and the CNN-classifier  are the heart of the machine, the scanner is just its 'arm'. They are both described, I hope better now, in the text

3. The literature overview and the current state of knowledge about ice crystal habits is not enough documented and detailed in the introduction and more in general in the text. See minor remarks below. I would like to see the instrument and its capabilities better put into the context of current state of research and the known research gaps.

I added a long text in the introduction, trying to explain how this type of instrument can be useful and interact with other techniques (CloudSat, etc) , on the plateau

4. Data access. The instrument is collecting data since 2014. So there are now 7 or 8 years of unique data collected at the Concordia station and not yet available. The data are the real added value of each innovative instrument. It is crucial in my opinion that as much data as as possible is made available, behind a DOI to allow for citation and recognition, at the time this paper will be published. I see it as a necessary complement to allow the readers to understand the quality and the scientific impact of the data produced by the instrument. Platforms as Zenodo, just to cite a common choice, are easy and user friendly and they allow to obtain a DOI and a citation for the dataset.

I welcomed   the suggestion: I submitted to Zenodo the CNN and its training database. The ICECAMERA database is the core of a second paper to be submitted in Copernicus. The new paper involves other instruments and other authors, and therefore I prefer to publish not the entire database, for the moment.

Minor remarks

1. In the introduction or somewhere in the text I would recommend to take the time to explain all the difficulties and possible failures associated to the long term operation of an instrument in such an environment. This is briefly mentioned through the text but I am convinced that the average reader would not get the idea of the actual and maybe less known challenges: breaking of rubber parts, (low) humidity issues in heated environments, etc.

    I added a section about these issues at the end of the paper

2. L14-19, L22-24: some references should be added to backup these scientific statements.

I did my best

2.  L29: 90 um ? 90 µm. Same through the text
    corrected

4. L34: change forms to shapes
corrected

5. L79: I believe the scanning principle should be briefly introduced before this point. Here we read after an entire scan, but we actually do not know yet what a scan exactly is, for this instrument.

I added some sentences and a reference

6. Fig. 3: I found this schematics not really informative. I would recommend either to use pictures of the actual parts of the instrument (a qualitative but illustrative way) or to provide a higher detail electrical/ communication schematics, even as Appendix material.

In fact, I changed the figure with a more communicative one. I realize in this moment  the quality of the picture is not the best, I'll send a better quality copy after acceptance of the paper

7. L139: what is the expected lifetime of the LEDs?

Sorry I have no answer for this question, the manufacturer is not explicit in this sense

8. L162: add a reference or a statement about the expected maximum height of blowing snow in Concordia.

added

9. L177 (and related section):

• Could you show data evidence of partial sublimation bias in the collected images? A caveat that you mention also later on in the manuscript.

• The role of relative humidity should be discussed here. If relative humidity is higher, sublimation rate will be lower.

Correct at 'high' air temperatures (-30°C). At low temperatures (-70°C) RH is secondary. This discussion was added in the text

10. Fig 8: I see some clear patterns / clusters / populations in the data. I recommend to try to explain why such patterns appear. I suspect they are related to individual events/seasons, but they may be related to more interesting physical aspects or technical issues. Also, I would try to produce similar scatterplots but including relative humidity, which should be routinely available in Concordia.

Fig.8 was deleted because it was confusing, and I could not analyze it statistically because housekeeping data were non collected continuously (g.e. they were not collected during the scan).

11. L294 (and related section): a flow-chart could help to present the processing steps rather than a bullet list. The text of this section should be edited to be descriptive rather than a simple enumeration of steps.

I added a specific flow-chart figure

12. L331: how often does this situation occur?

Added a sentence

13. L352: Why this approach end up to be unreliable for your instrument?

I didn't comment this because using only shape factors for crystal classification was just a mess, not only for ice-camera, but for science in general. Fortunately ,CNN were invented…

14. L363: does this imply that the images collected have to be resized?(down or upscaled?) Is the resolution of the instrument constant?

Yes, ice-crystal images are re-scaled but only for the CNN classifier. Measurements are taken on the original image

15. Fig 14: can spheroidal particles actually be supercooled liquid water? It would be good to have some size reference in this image.

I cannot distinguish solid from liquid, but the presence of supercooled water is increasing in Concordia, some collagues are publishing about this. I don't know the fate of a supercooled droplet when she hits the window of icecamera. I don't know if freezes immediately ..like on our mountain trees, and I cannot test this

16. L402: about trigonal plates. This statement here seems to be contradicted by the results of Fig. 17, and in fact this is highlighted in L471.I would either remove the statement here or remove this class from all the analysis of the paper.

I changed the training dataset for the (new) CNN and now trigonal plates are finally detected

17. L407: a good idea actually.

Thanks!

18. L560-561: I maybe missed it, but are there statistics on the total amount of data collected since the instrument is operational?

I added a figure with an overview of the number of collected particles from 2004 to 2021

19. L565: I would recommend to clarify explicitly the content of the governmental policies mentioned here. Nowadays most of the institutions promote FAIR policies (https://www.go-fair.org/fair-principles/), so thisCode and Data availability statement may be sounding strange for some readers. Secondly, only the CNN and a test dataset is mentioned here.The paper, as the title suggests, is not about the CNN but about the instrument. I would expect to see a data statement about access to the data collected so far by the instruments and maybe some link to relevant technical documentation of the instrument itself.

I finally published in Zenodo the CNN and the ice-camera dataset used for the training+validation+test of it. The ref. is in the text of the new verion of the paper. As the topic of the paper is the instrument and its software, I don't release for the moment the entire data set. The data set is going to be used in a second Copernicus  paper with other co-authors and other instruments involved

20. Plot quality: the quality of some plots seems very low (for exampleFig. 5)

The figure 5was refurbished. Most figures are vector figures (MATLAB) and can be modified. I realize in this moment that fig.3 is still low quality, I'll produce it in high resolution as soon as the paper will be accepted.

%%%%%%%%%%%%%%%%%%%%%%%%%%%%%%%%%%%%%%%%%%%%%%%%%%%%%%%%%%%%%%%%%%%%%%%%%%%%%%%%%%%%%%%%%%%%%%%%%

**REFEREE 2**

1)Abstract is not adequate.

I sligthly Changed the abstract

2) L77-78: "The principle is simple: at low temperatures and low wind speeds (conditions encountered at DC), precipitation falling over a flat glass accumulates with time and remains frozen until it sublimates, …"

Language should be improved, but also the collection and issues with collection efficiency should be discussed here or later in the paper. Particles deposit on the DS, but what makes them stay there ("remains frozen")? What are the effects of wind both during collection (deposition) and after collection (can particles be removed by wind)? Could any biases be introduced based on wind effects or other effects depending on size or shape?

 A discussion about 'what makes them stay there ' was added. I  added  a discussion about the  wind-speed threshold for wind drift (and thus for the removal of particles from the DS) based on specific Dome-C results. I Added a discussion on the viscous boundary-layer on the surface of icecamera

3)1.2 Camera/1.3 Focusing and 2.1.2 Limitations:

Say something about resolution (resolving power). Pixel size corresponds to 7 um, but resolution does not seem to allow detecting particles smaller than 70um (70um, not 60um as you state in L333, is equivalent to 40000um2).

Corrected

How much does the focus change between calibrations? How long does focus calibration take? (Could it be done before each scan instead of 6h intervals?)

Corrected:  The typical focus adjustment between two calibration if between 0 and ±0.25 mm. The calibration takes approximately 5 minutes and for this reason it is not performed after each measurement in order to spare PC resources for data processing.

Consider including a discussion of accuracy of provided particle sizes.

A sentence about the calibration of the actual pixel size on the images was added

4)Thermal control:

L168-170: "The NI-Labview software controls the internal temperature of ICE-CAMERA above -40°C, and the DS temperature always under -5°C ("Heater 1" in Fig.3)."

Is this always possible?

Specify if this is general heating of ICE CAMERA or specific for sublimation (then refer to Sect 1.6.1).

**Corrected:** These conditions are maintained all-year round in all phases of the measurement cycle

Unclear where Heater 1 and 2 are.

Heater 1 and 2 swapped in Fig 3.

Corrected

L179-181: "An indoor test (Fig.7), showed a heating of rate of 2.5°C min-1, and a cooling rate of 1 °C min-1. The cooling rate is almost 50% of the heating rate just due to the sandwich heating-glass structure, with the heating layer at the middle"

"…just due to the sandwich...": is this passive cooling, what is "heating layer at the middle"?

Corrected: I also introduced a description of the 'sandwich' and its thermal characteristics

Is the indoor test really relevant here? You then report the heating/cooling rates outside. So, Fig7 could maybe be replaced with a cycle outside.

I prefer to keep this picture, as in steady-air the same thermal cycle (with just a different offset temperature) occurs outside, as the thermal constants of the instrument are always the same.

In Sect 1.5 you mentioned about heated air between glass sheets (DS and second sheet) and keeping DS frost free; is that the actual heating of the DS?

I Tried to clarify this point

Detail: how much air pumped inside and through the space between the windows?

Done: The outside air is pumped (through a 3.5 l/m miniature pump) for five minutes every hour

Distance between glass sheets?

Corrected: 13 mm

L192: "Heating is anyway interrupted if the DS temperature exceeds -5°C…"

How often does this happen?

Only in warmest summer days

5)Deposition/scan/sublimation cycle:

 The cycle of deposition, scan, and sublimation is not properly explained in detail. The reader gets the details somewhere between the lines.

Corrected: better described the cycle

L190-192: "…sublimation of the majority of particles (D<1000 um) is complete within 20 minutes, with just a few big (D>>1 000 um) grains still present after 30 minutes.

After these tests, the heating time was set at 10 min."

"D" is not defined.

Corrected

"within 20 minutes": 20min after the 10 min heating period (after 10min heating was turned off)?

Corrected

"few big grains" Can you be more specific? After 30 min grains were larger than 1mm? Or ice particles that original had D larger than about XY mm?

Corrected: (initial diameter>>1000 um)

Then you continue to discuss the cycle and for how long the DS is "sensitive to falling ice particles" and that this time is variable depending on conditions and that there is an uncertainty associated with this.

What is in L195 the "evaporative removal of particles during the accumulation period"? Is this wind related; why evaporative now and not related to sublimation; what is accumulation period".

corrected

All these details should be explained more clearly and then the issues discussed clearly too. In particular, make it clear that the deposition phase is not clearly defined (only its end due to the scan). I think I have not seen how long a scan takes. Would scanning more often be a way to study sublimation issues related to smaller particles (100 um)? Would could then see how they sublimate in consecutive scans.

Corrected:
evaporative→sublimation
accumulation period→ 'Deposition period' (now defined)
Scan duration-> 2 minutes

-Would scanning more often be a way to study sublimation issues related to smaller particles (100 um)? Would could then see how they sublimate in consecutive scans.

This is possible, and can be planned (in summer), as a test ,on a few target particles, but in this specific instrument it cannot be done (and was not done) routinely, because the image

processing of a typical population of hundreds ice particles takes 10 minutes or more. Small DD particles such as small (<100um) plates, the most interesting in a sublimation test, are very scarce in summer. Moreover, in winter the instrument works unattended, and experiments of this type are virtually impossible.

6)Fig 8:

What measurements does the figure show? For one 1h cycle, how many temperature measurements are shown?

Fig 8 left) at what wind speed?

+25degC at still air, +20degC at 8m/s: are those temperatures averages, median, or ? (I can see a large range, maybe 20degC to 35degC.)

Wide spread in blue; two regions (sublim and deposition phases)?

Left: better also show dT instead of DS temperature.

Could Fig8 be separated in one fig. only the deposition periods of 20 min prior to scan, one for after/during sublimation heating? This could be clearer instead of heaving heated and not heated temperatures mixed in one figure.

The figure was in fact confusing, and was totally removed

7)Sect 1.6.2 Sublimation of ice particles

You discuss sublimation, both unwanted and wanted sublimation here. When saying "negative effect" you refer to the unwanted sublimation. Be careful to keep clarity in this section.

Corrected

L214: "as DS ice is always super-saturated relative to the surrounding air"

Wrong as it is stated. It is also unclear what "DS ice" refers to.

At the heated DS the equil. vap. pressure will be higher; thus the surrounding air will have a lower RH at the heated DS and ice on the DS will experience sub-saturated conditions.

L215-216: "the vapour pressure of ice on the DS relative to the surrounding air (saturated relative to ice…"

Wrongly or unclearly stated: "the vapour pressure of ice on the DS..." should be something like:

"the vapour pressure saturated relative to ice at the DS temperature".

Fig 9: Why show this ratio (e0ice(T_DS)/e0ice(T_amb))?

Removed the discussion above, as it was confusing. Thanks!

More intersting would be the inverse (e0ice(T_amb)/e0ice(T_DS)), i.e. RHice at DS if surrounding air is at RHice=100%. For dT=5K this would range between 45% at -80degC to 65% at -10degC. It shows the sub-saturation. Fig 9 is not further discussed or used later on. If you want to keep it, I suggest to plot inverse.

L214…..==Fig.9 and its discussion were eliminated, as unnecessary: the simulations of sublimation contain  the same information and lead to more readable results==.

You use over saturation, oversaturation, over-saturation, and super-saturation. Only use one term for clarity, I would suggest super-saturation.

==Corrected ,adopting super-saturation==

L228: "growth rate of the facial area" Growth perpendicular to facets?

==Corrected==

L229-230: Unclear what critical super-saturation and critical temperature are.

==Sentence removed==

L231 "0.05% to 5% under-saturation". How is "under-saturation" defined here?

==Corrected in 'sub-saturation'==

L234 "The steady-state shape of the sublimating crystal depends"

Unless explained or defined, it seems contradicting: a sublimating particle is not in steady state.

==Text was modified and 'steady-state shape' removed==

L246-247 "The simulations assume the completion of preliminary sublimation of points and edges of the particle"

Unclear language (preliminary, points, edges).

==Change==d: ==The simulations assume that the preliminary sublimation of  the high-curvature parts of the particle was already completed==

L252 "still evaporate" Unclear what the "still" refers to => "sublimate".

==Corrected==

L253 "could survive along the heating period" => "will survive longer than the heating period"?

==corrected==

L259: Meaning of "After the sublimation period, DS is exposed to falling crystals."?

DS is always exposed. Particles depositing during the sublimation period may partially sublimate and then be included in "collected" particles. (see comments for Deposition/scan/sublimation cycle above)

Corrected:During the deposition period, the collected particles also undergo sublimation, although this is much slower than during the sublimation period.

Fig11: Do you have any experimental data on sublimation times for actual particles (similar to your spheroids)? That would help to set your statement in L270 in relation, or that you only see few particles with partial sublimation.

This experiment is possible, and can be planned (in summer) on a few particles, but in this specific instrument cannot be done routinely. In winter, the instrument works unattended and experiments of this type are almost impossible. Moreover, small DD particles such as small (<100um) plates are very scarce in summer.

8)Sect. 2.1.2

1) Specify better: 2000 per scan?
yes, corrected

2) Explain what you mean with "By default". Relate to resolution (see 3) above).

Corrected

3) Specify the segmentation error (particles are overlapping?). How can a particle be counted twice (double counting)? What is 12% in your worst case (overlapping particles account for 12% of particles in one scan?)?

Corrected : 12% of particles in one scan. More details were added

4) Two issues are brought up together but are rather two limitations that should be listed/discussed separately. It is unclear how and on what basis these spheroidal particles are disregarded.

Corrected

9)Sect 2.2.2 Training dataset

I would like to see a better description of the various datasets.

What is the "image dataset" (L437)? How many images; from what time period?

Changed: period 2014-2017. How have these images been selected out of all available images? Randomly or by some other selection criteria?

Changed the sentence

This image dataset is then apparently split in 10% validation dataset and 90% training dataset (25705 images).

What about a testing dataset?

Say how testing (see Sect 3) was performed, in Sect 2.2.2 or Sect 3 for example. How many and which particles were used to test the CNN after training is complete, i.e. to produce the results shown in Sect 3?

You are perfectly right: for the testing I used a testing data set including approximately 50% of training images and 50% of unused images. This 'error' was corrected in the new version: now 80% of the whole dataset used for the CNN training was used as training data set, 10% for validation data set, 10% for testing data set. Moreover, I increased the number of training particles for most classes. Trigonal plates are rare, and their number in the training dataset was thus artificially augmented by duplicating the training images for this class, in order to avoid their absence inside most of the small (64-images) training mini-batches. Image augmentation during training contributes to a partial spatial randomization of this special class.

I added information about training, testing and validating data sets. The training dataset, together with the CNN, was published on Zenodo.

Provide more details about the augmentation: To what size is the original training dataset of 25705 images increased after augmentation? Each class augmented the same (number of images increased by same factor)?

There is no formal answer to this question: Image augmentation is introduced randomly (within the fixed limits and transformation types) to each image of each mini-batch used during the CNN training. Each mini-batch represents a random sample of the data base.

10)Biases during training

 The number in each class is different. This can generate biases in the CNN, please comment. In particular, I think that part of the low performance for trig plates is due to the fact that this is the smallest class in the training dataset.

I increased the number of training particles for most classes in order to better equalize their numbers. Trigonal plates are rare, and their number in the training dataset was thus artificially augmented by duplicating the training images for this class, in order to avoid their absence inside the small (64-images) training mini-batches. Image augmentation during training contributes to a partial spatial randomization of this special class. The resulting CNN is more performant, with 90% classification accuracy of the training data set just after 3 epochs, compared with 83% (after 5 epochs) of the first version of the paper. The accuracy in the detection of trigonal plates is now reasonable 86% (on training data set).

11)Data

A few example images in the full resolution should be included in the paper. This will help when discussing resolution and sizing accuracy (see Major point 3) above).

Fig.16 already shows examples of good quality ICE-CAMERA images

Reconsider if you interpreted government policies with respect to sharing data correctly. It would be useful to share the whole image dataset. Should that not be possible, then please share at least a sample dataset.

The training dataset, together with the CNN, was published on Zenodo.

.

Minor comments

==================

Consistency with terminology and spelling in various places.

E.g.

GoogLeNet, Dataset

Corrected both

I would expect that Introduction is Sect 1.

Sect 2 is on instrument; 2.1 Overview (instead of 1.1)....

Corrected

L69: "In this work, the term 'precipitation' will include both diamond dust and"

Should perhaps refer to "precipitating diamond dust".

Corrected

"cooling speed" => "cooling rate"

Corrected

"eventual" => "occasional" (2 x)

Corrected

"deposition window" => "DS"

Corrected

"By the way" is not good English for an article.

corrected

Fig 12: "Reg.growed" Use correct label and be consistent with text.

corrected

Fig 13: Remove title "ICE-CAMERA: Summary of detected grains…"

corrected

Be consistent: what is synthesis image?

summary-image,synthesis image,mosaic, summary image

corrected

L325 "measures" is not a noun.

corrected

L325 and L83: specify what weather data are.

Added a sentence

L384 "hexagonal prism"

I would call this class "long column" or something else with "column"

Fig 14 "compact prism" should be "compact column" (as in L395).

corrected

L401: Why refer to "insoluble"?

corrected

L451 "3.1 Precision of the classifier" should be "3.1 Accuracy of the classifier"

corrected

L460 "mistaken" = "mistaken with each other"

corrected

L 462 "Compact columns are misclassified almost 20%"

I read 15.4% in the bottom row, not 20%.

A new discussion about the (updated)  CNN results  was written

L473-474 "The three-dimensional structure of the ice particles is lost in the ICE-CAMERA images, so that some thick ice forms such as C4a, P1b, G3b, CP1a, etc. (Kikuchi et al, 2013), if any, are likely to be misclassified"

Be a bit more specific (or give at least one example).

is quite complicated for me to find 'clear' examples in my database

L498-499

"The classifier was used…" You are showing results of the classifier but then also of the "image measurements" for the resulting classes.

corrected

Is the Jan-Feb 2017 period not part of the image dataset used for training/validation and testing?

The image data set used for training/validation includes images from 2014 to 2017 (so, also jan-feb 2017). But just a random sample of the whole image data set was used for training/testing, and only a few images from the Jan-Feb 2017 period were thus used in training/validation

L506 "maximum whisker length is here equal to the interquartile" seems wrong.

Total box height is equal to the interquartile range. What do the whisker lengths indicate?

Corrected, my mistake! Lower and upper whiskers mark one interquartile below the 25th percentile and one interquartile above the 75th percentile, respectively

L507 "some relevant differences" seems to refer to one relevant difference.

corrected

L532 "diameter of the circle equivalent to the bounding box"

definition is ambiguous:

"diameter of the circle with the same area as the bounding box

Similarly, in L533: ambiguous definition.

corrected

Fig 23: Suggestion: put (as whiskers only) on scatter plot Feret-box vs projected-surface equiv. diameters. In that way direct visual comparison is possible.

done

L543-544 "Commercial or customized instruments do not have this flexibility, more typical of old-style handcrafted products"

Purpose of this comparison, what instruments are you comparing to?

removed

L548 "convolutive" => "convolutional"

corrected

L552 "precisely" Do you mean precise here, or accurate?

Anyway, neither of the two was discussed.

removed

See Major point 3) above.

Concluding remarks on potential improvements (in particular with respect to the sublimation problem). Use of ICE-CAMERA in other environments (not Antarctica)?

added in the conclusions

---

## Referee Report (RR1)

amt-2022-62
Review after revision 2

Fig. numbers, Sect.s, and Lines refer to amt-2022-62-ATC3.pdf

Comments to author:

I appreciate the corrections and modifications done to improve the manuscript. However, I still see several small issues, which, together, still result in some confusion, ambiguity, and potentially wrong information or statements.

**Double window**
In Fig.3 "double window" seems to refer to the double window interspace (what you call "double window space" or "double window inter-space").
In L. 193 "double window" has not been mentioned or defined earlier.

**DS**
There is a potential confusion because "DS" refers both to a surface (e.g. in Fig. 3 or when talking about the DS temperature Ts) as well as to the sandwich of three glass layers. When "DS" is defined (L.200) it refers to the sandwich of three glass layers. This "DS" or sandwich is also referred to as "heated glass" (e.g. Fig. 3 or L.231), "electrically-heated glass" (L.200), "heating-glass" (L.250).

**Heating and thermal control**
- Instead of providing the specific heating power of 1000 W m-2 (L. 200/201), I would provide the actual heating power (in W), or provide both.
- The "heated glass" is used to achieve sublimation (L. 202). It is unclear if the "DS temperature of (at least) 3 °C above air temperature" (L. 205) refers to ICE-CAMERA and if this is temperature controlled using the DS "heated glass". If yes, clarify in Sect.2.6. Do you consider 3 °C or 5 °C?
- What is a "ventilated resistance" (L. 230)?
- L. 231, *"eventually"* is the wrong word: *"disabling… during sublimation (see Sect. 2.7) when needed"*.
- L. 232: *"200W wired thermostat"*, I am unsure what the "wired" refers to.

**Fig. 8**
If you want to keep Fig. 8, then several points should be addressed:

- "air" and "air" should refer to the outside air and the air in the double window interspace, respectively; change labels accordingly. Consequently, "Tair" is not the same as "Tair".
- Similarly, "Ts" is not the same as "Ts", one is the DS temperature, the other Ts is the temperature on the inside surface of the sandwich of three glass layers (surface towards the double window interspace).
- Radiative and convective/wind effects only apply to "k2" of the outside air. For explaining the cooling in principle, it may be sufficient to consider the outside-air k2; i.e. you may remove k2 of the double-window interspace.

**Explanation of sublimation and deposition periods**

Consider rewriting L273-274 *("Once the heater is turned off, and after a cooling time of 20 minutes, the DS temperature comes back warmer than the air by about dT = 5°C. The "sublimation period" is considered complete, and ice particles accumulate again on the DS, with no relevant sublimation ("deposition period" in Fig. 7).")* more clearly, e.g:

*"Once the heater is turned off, and after a cooling time of APPROXIMATELY 20 minutes, the DS temperature comes back TO BE warmer than the air by ONLY dT = 5°C. AT THIS POINT, THE "sublimation period" (OF APPROXIMATELY 30 MINUTES) is considered complete, and ice particles START ACCUMULATING again on the DS, with no relevant sublimation, I.E. THE "deposition period" STARTS."*

Note that Fig 7 refers to an indoor test, but the text above to general outdoor behaviour (or an outdoor test) not shown in any figure. Thus, be careful when referring to Fig.7 to do it in an adequate way.

Note also that in Fig 7, "deposition" and "sublimation" do not bear a physical meaning.

**Re: C) L 274-292 Adhesion of ice on DS**

The two examples you provide in the explanations in L.291-295 give adhesion speeds (particles below these limits adhere) that are smaller than the settling speeds. Yet, you claim that this is sufficient to explain adhesion of these smaller particles. My interpretation of these explanations is that particles settle at speeds faster than adhesion speeds and therefore should not adhere (without considering other effects).

Perhaps I am missing something or mis-interpret. In any case, it would be good to improve clarity.

In L.291, "*decreases with particle diameter*" leaves me wondering if you mean "*decreases with DECREASING particle diameter*".

The "*<>*" in L. 298 should perhaps better be called *"large"*.

**Diameters**

L.460 "Particles below 3600 µm2 in bounding-box surface, 73 pixels minimum size (equivalent to approximately D<60µm)":

- "D" is ambiguous, use Ds or Df (assuming a spherical particle, I think you want to refer to Ds, which would be equal to Feret width and Feret length);
- "73 pixels minimum size": I would avoid "size" as you seem to refer to an area of 73 pixels;
- It is unclear what the limit for processing is, 3600um2 bounding box or 73 pixels area?

L. 449 In one occasion you use "Feret size": should be Feret width and length?

L420-422 define Df and Ds, where Df is the "Feret-box surface-equivalent diameter". Later , in L.715 you talk about "the bounding box equivalent diameter (Df)"; I would be consistent.

L.716-717 "For comparison, a round particle is expected to have a ratio of Ds/Df=0.78.": SHOULD BE (accord. to def. in L420-422) sqrt(pi/4), approx. 0.89, and not 0.78!?

**Overlapping particles**
Sect 4.1.4, 3): Could these artificially created copies of the "same segmented image" be detected and thus avoided as an issue?

**Positive bias of confusion matrices**
L. 545-550: The text modified in response to my previous review is unclear and leaves my comment not properly discussed. It is unclear what "This process has been repeated three times recursively…" means exactly. Is the enlarged training datatset used to create a new CNN, which is then used to classify the previously discarded images (wrongly classified by previous CNN version)? It seems that this enlarged "image training data set" is used for consecutive training and testing. Data used for testing should be independent from training, but here it seems that the test data (or a large part of them) were part of training the CNN (or classified by a previous version). I see the risk of introducing a positive bias by effectively preventing (to a large extent) data that may be classified wrongly by the trained CNN because wrongly classified particles are excluded in this process of enlarging the training and test dataset. I don't see how *"these images should be considered as ordinary, supervised training images"* adequately discusses this.

**Discussion about importance of sublimation (Re comment on L624/5, 632/633 in previous review)**
L647-649, "This result suggests that sublimation on the DS during the deposition period is less important than the natural variability of precipitation intensity in determining the number of particles detected during the acquisition.": I am not sure that you can draw a conclusion like this based on the evidence you are mentioning. In addition, the following sentence, which should confirm this, is confusing as you seem to refer to lower (colder temperatures) as "temperatures above the median" and warmer temperatures as "in the lower temperature range".

Also L 654-655 suggests that "sublimation does not affect dramatically the number of particles". Again, this cannot be concluded from the results. Contrarily, one must expect that sublimation affects the number of particles (see Fig.9b in Sect 3.2) for smaller particles and at warmer temperatures.

Similarly, the statement in L 691-692 cannot be concluded from the measurements. The resulting size distribution in Fig. 23 may very well be affected by sublimation (and would have been different, with larger and/or more particles, without sublimation). If plates were observed at -50degC or warmer (temperature information is missing in the discussion) then the observed plates may be deposited at the end of the deposition period, shortly before the scan. Particles deposited earlier would have been completely sublimated if less than 200um in size (see Sect 3.2).

Likely, the discussion cannot be improved to support the author's claims. Rather, it should be acknowledged that sublimation cannot be excluded or quantified easily. In addition to particles potentially disappearing due to sublimation (L731-732), particles may also appear smaller than their size during precipitation and when deposited. Statements such as in L738 ("apparently without dramatic losses of small particles for sublimation") cannot be deduced from the presented observations.

**Other issues:**

**Resolution of the ICE-CAMERA**
Your clarifications refer to the pixel size in the scanned image, but not to the optical resolution. Can anything be said about the optical resolution? (What are the smallest details/features that can be seen on the images?)

**Use of "adverse"**
L 307, "The adverse effect is an accelerated natural sublimation of deposited particles." As in the previous review, I have a problem with this sentence and think it is bad language and therefore difficult to follow.

**Section numbering**
Sect. 4.1.1 is missing?

**Eventually**
L 628, "eventually" is the wrong word.

**Data set or dataset**
You use "data set", data-set", and "dataset". In one place also "data store", unclear if you mean dataset there or something else.

---

## Author Response (AR2)

REFEREE 1

- L. 21: Hydrometeor larger than 10 mm can be found near the coast, mostly as aggregates (some can be found in the dataset: https://doi.org/10.1038/s41597-022-01269-7)

Added

- L. 45: another examples for the usage of disdrometers is https://doi.org/10.1016/j.atmosres.2017.06.001

Added

- L. 162: how often this is done? please clarify already here in the text.

done

- L. 214: if there is wind-drifted snow (and so strong enough wind), how can the author be sure of what is stated in L. 120 ? "precipitation falling on a horizontal glass surface accumulates with time until it sublimates, leaving plenty of time for scanning...")

I changed the sentence

- Fig. 8: should this figure also illustrate some temperature profiles? Additionally, please explain in the figure or in the caption, not only in the text, all the short variable names used.

done

- Fig 22, 23, 24: relevant figures. As there are many other figures, maybe they could be combined in a multi-panel image (same for otehher figures before). Additionally, maybe some information is missing to better interpret the meaning of such figures: for eyample the total number of scans of each histogram and the underlying distribution of relative humidity, wind, temperature. The goal would be to see if the distribution appearing in these histograms are due to characteristics of the instrument at various environmental conditions or are just the representation of the distribution of the environmental conditions themselves.

I tried to bind together many figures . I also added the histograms of Temperature, RH, wind speed for the measurement period, as suggested

- About the very large number of figures: please consider multi-panels and also one or more Appendix sections if relevant.

I tried to bind together many figures ..

REFEREE 2

The manuscript has improved significantly with the modifications and additions by the author. Nevertheless, some unclarities remain, which I am pointing out below. In addition, I am listing a few minor things and oversights. With Fig. numbers, Sect.s, and Lines I am referring to amt-2022-62-ATC2.

A) The heating design is still unclear.
1) I appreciate the clarification about the structure of the heated-glass DS. This is what the author seems to call sandwich-like structure. I had confused that with the "double window" consisting of DS and at 13mm distance a second glass sheet with heated air in between, not being aware of that the DS itself has two glass layers with a heating layer in between. The detail that is provided in Fig. 8, could be included as text in Sect 2.5. Fig. 8 is not needed and should be removed.

I tried to better explain the thermal design following the indications of ref.2. Only, I kept Fig.8 , as I personally think it's useful. I added to it a sketch of the temperature profile (as suggested by ref.1)

2) The sentence referring to Fig. 8 in Lines 245/6 is still confusing or misleading due to a seemingly wrong use of "almost". I would suggest to rephrase it to something like:
"The cooling rate is at most only about 50% of the heating rate. The cooling is passive by heat transfer to the surrounding air."

done

3) The new paragraph in L 250-253 is not needed. While it is true, the actual difference between cooling and heating rates depends, in addition to the mentioned heat transfer coefficients, on the heating power of the electrically heated DS glass.

I changed a sentence, but kept the paragraph

4) Heater 1 is a thermostat-controlled heater. I would change, in Fig. 3, "GAS TERMOSTAT" to "THERMOSTAT".

done

5) I am still unsure about Heater 2. Is it the electrically-heated DS glass? Or is it a heater working together with a pump blowing air between the DS glass and the second glass sheet 13mm below the DS. In the previous MS version it said "air is pumped…, filtered, heated and finally blown through the double window space…". Or is it yet another heater? In this respect, the sentence in L 227-229 is unclear. Is it Heater 2 controlling the inside temperature to be above (warmer) than -40degC? What is controlling the DS temperature to not rise above (warmer) than -5degC (cannot be a heater, but the author refers to Heater 2)?

I changed the sentence, I hope it is better written, now

6) L. 235 "…Mylar… prevents overheating of the DS above -5degC". I would change that to be more specific and correct (and connect to the next sentence): "…Mylar… prevents overheating (of the instrument) and allows keeping the DS below -5degC… . Additionally, in warm weather… ".

done

B) The resolution of ICE-CAMERA is not given. From the pixel size of 7 um and a 1:1 lens I am assuming that 1 pixel corresponds to 7um (pixel resolution). The author mentions "a fine calibration" without specifying the result of it (confirm 7.0 um/px?). The sentence in L 157 says something vague about the potential resolution of the macro lens, but not of the actual optical resolution of ICE-CAMERA. What are the smallest details/features that can be seen on the images? I doubt that details of 7um size can be seen if one pixel is 7um.

I Added a sentence with the fine calibration details

C) L 274-292 Adhesion of ice on DS: I am a bit confused. It seems you show that particles at normal speeds below their settling speeds would adhere. I.e. if they settle at settling speed they may not adhere.
Is there a limit of wind speed of 5 m/s imposed (mentioned in L 643) that has to do with adhesion?

I didn't change too much this part, because in the text I already explained that small particles can stick for van Der Waals forces alone, while big particles stay attached for electrostatic forces as van der Waals forces are quite weak and not sufficient alone to stick on the DS the 'big' particle in free fall

D) The total number of particles is 81800. Of these only 5500 have been manually classified. All other images have gone through CNN classification followed by manual verification/correction (as explained in L 525-532). That means that many of the around 8000 images (10%) used for testing and compiling the confusion matrices have already been classified by CNN. Does this introduce a bias (improving accuracy in the confusion matrices)?

I Added a discussion

E) L 610 suggests that many particles are rejected for statistical analysis. It would be interesting to know roughly how many and for what reasons. Major reason wind speed threshold? Or too small particles?

I Added a discussion

Other minor things:

L. 54: Consider specifying why "In Antarctica, their (MASCS) practical application is mainly limited to coastal areas."
I added a sentence

The titles of Sect.s 2, 3, 4, and 6 are missing. Additionally, in the numbering Sect 5 is missing completely and Sect.s 6 appear twice.

Done, thank you!

L 158: "A 90 deg bending aluminium mirror…" seems to refer to the "45deg mirror mentioned earlier, I suggest again to use same terminology and names as before: "The 45deg mirror…".
Now "DS" is not introduced when this term is first used in Sect 2.1. In addition, in some of the added text there appears "SD" instead of "DS" (L.s 126, 221, 262, 289, 337, 624).

Done

L 211: 3.5 l/m => 3.5 l min-1 or 3.5 l min-1

Done

L 261 galss => glass

Done

L 271: "The non-contact…" => "A non-contact…" (talking about something not mentioned before)

Done

L 297: "The adverse effect is an accelerated…" => "An adverse effect is, however, an accelerated…" (or "A disadvantage…")

Done

In Sect 4.1.4, "1)" is missing in the numbering.

Done

In Sect. 4.1.4 4): Unclear how overlapping particles can be counted twice. Contrarily, two overlapping particles would be counted only as one (cluster). Similarly, merging close-by-particles (point 5) due to region growing results in less counts.

I added a discussion

L 560: remaining 80% of the training => remaining 80% for the training

Done

L 605: 8% regular plates => 9% hex. Plates

Done

L 611: "errors in … contributed to…" => "problems with … resulted in…"

Done

Sentence in L 624/5 suggests that (it is clear that) sublimation on DS is less important than natural variation. This is not clear or obvious to me. Perhaps give a hint why or how. Similar in L 632/3.

I tried to better discuss this point, adding (as suggested by ref.1) the hystograms of T,RH,Wind speed for the measurement period (2014-2021) for comparison with NpS histograms

L 631 HR => RH

Done

---

## Author Response (AR3)

Follow, in yellow, my point-to-point answers to the comments of referee 2.

 (Line numbers of the new version  do not correspond with those in the  Referee 2 comments)

%%%%%%%%%%%%%%%%%%%%%%%%%%%%%%%%%%%%%%%%%%%%

Double window

In Fig.3 "double window" seems to refer to the double window interspace (what you call

"double window space" or "double window inter-space").

In L. 193 "double window" has not been mentioned or defined earlier.

DS

There is a potential confusion because "DS" refers both to a surface (e.g. in Fig. 3 or when

talking about the DS temperature Ts) as well as to the sandwich of three glass layers. When

"DS" is defined (L.200) it refers to the sandwich of three glass layers. This "DS" or sandwich is

also referred to as "heated glass" (e.g. Fig. 3 or L.231), "electrically-heated glass" (L.200),

"heating-glass" (L.250).

Throughout the updated text and figures, I evidenced that DS is just  the outer surface of the electrically heated glass. I supposed it was straightforward also in the previous version, but I hope now is much clearer.

Heating and thermal control

• Instead of providing the specific heating power of 1000 W m-2 (L. 200/201), I would

provide the actual heating power (in W), or provide both.

Done

• The "heated glass" is used to achieve sublimation (L. 202). It is unclear if the "DS

temperature of (at least) 3 °C above air temperature" (L. 205) refers to ICE-CAMERA and

if this is temperature controlled using the DS "heated glass". If yes, clarify in Sect.2.6. Do

you consider 3 °C or 5 °C?

3°C refer to a referenced paper carried out in DC, as stated in the text, in ICE-CAMERA we keep 4-5°C above air T

• What is a "ventilated resistance" (L. 230)?

Changed into 'ventilated heater'

• L. 231, "eventually" is the wrong word: "disabling… during sublimation (see Sect. 2.7)

when needed".

changed

• L. 232: "200W wired thermostat", I am unsure what the "wired" refers to.

Removed 'wired'

Fig. 8

If you want to keep Fig. 8, then several points should be addressed:

• "air" and "air" should refer to the outside air and the air in the double window interspace, respectively; change labels accordingly. Consequently, "Tair" is not the same as "Tair".

• Similarly, "Ts" is not the same as "Ts", one is the DS temperature, the other Ts is the temperature on the inside surface of the sandwich of three glass layers (surface towards the double window interspace).

• Radiative and convective/wind effects only apply to "k2" of the outside air. For explaining the cooling in principle, it may be sufficient to consider the outside-air k2; i.e. you may remove k2 of the double-window interspace.

Removed the fig.8 and summarized the discussion.

Explanation of sublimation and deposition periods

Consider rewriting L273-274 ("Once the heater is turned off, and after a cooling time of 20 minutes, the DS temperature comes back warmer than the air by about dT = 5°C. The "sublimation period" is considered complete, and ice particles accumulate again on the DS, with no relevant sublimation ("deposition period" in Fig. 7).") more clearly, e.g:

"Once the heater is turned off, and after a cooling time of APPROXIMATELY 20 minutes, the DS temperature comes back TO BE warmer than the air by ONLY dT = 5°C. AT THIS POINT, THE "sublimation period" (OF APPROXIMATELY 30 MINUTES) is considered complete, and ice particles START ACCUMULATING again on the DS, with no relevant sublimation, I.E. THE "deposition period" STARTS."

Done, thank you!

Note that Fig 7 refers to an indoor test, but the text above to general outdoor behaviour (or an outdoor test) not shown in any figure. Thus, be careful when referring to Fig.7 to do it in an adequate way.

Note also that in Fig 7, "deposition" and "sublimation" do not bear a physical meaning.

text changed in order to meet these requests

Re: C) L 274-292 Adhesion of ice on DS

The two examples you provide in the explanations in L.291-295 give adhesion speeds (particles below these limits adhere) that are smaller than the settling speeds. Yet, you claim that this is

sufficient to explain adhesion of these smaller particles. My interpretation of these

explanations is that particles settle at speeds faster than adhesion speeds and therefore should

not adhere (without considering other effects).

Perhaps I am missing something or mis-interpret. In any case, it would be good to improve

clarity.

I made a matlab code using the theory developed by the refenced paper on snow attachment to the glass
and sedimentation speed papers (paper added to the references) , and  compared the sedimentation speed
of spherical particles with the sticking, critical speed. A discussion is added in order to clarify this topic.

In L.291, "decreases with particle diameter" leaves me wondering if you mean "decreases with

DECREASING particle diameter".

corrected

The "<>" in L. 298 should perhaps better be called "large".

Diameters

corrected

L.460 "Particles below 3600 µm2 in bounding-box surface, 73 pixels minimum size (equivalent

to approximately D<60µm)":

• "D" is ambiguous, use Ds or Df (assuming a spherical particle, I think you want to refer

to Ds, which would be equal to Feret width and Feret length);

corrected

• "73 pixels minimum size": I would avoid "size" as you seem to refer to an area of 73

pixels;

corrected

• It is unclear what the limit for processing is, 3600um2 bounding box or 73 pixels area?

corrected

L. 449 In one occasion you use "Feret size": should be Feret width and length?

Removed the ' Feret size' from  sentence

L420-422 define Df and Ds, where Df is the "Feret-box surface-equivalent diameter". Later , in

L.715 you talk about "the bounding box equivalent diameter (Df)"; I would be consistent.

corrected

L.716-717 "For comparison, a round particle is expected to have a ratio of Ds/Df=0.78.":

SHOULD BE (accord. to def. in L420-422) sqrt(pi/4), approx. 0.89, and not 0.78!?

Corrected the error, THANK YOU!

Overlapping particles

Sect 4.1.4, 3): Could these artificially created copies of the "same segmented image" be

detected and thus avoided as an issue?

Added a sentence for this. Also, a sentence about the duplicated detection of the same , non overlapping particles was added.

Positive bias of confusion matrices

L. 545-550: The text modified in response to my previous review is unclear and leaves my

comment not properly discussed. It is unclear what "This process has been repeated three

times recursively…" means exactly. Is the enlarged training datatset used to create a new CNN,

which is then used to classify the previously discarded images (wrongly classified by previous

CNN version)? It seems that this enlarged "image training data set" is used for consecutive

training and testing. Data used for testing should be independent from training, but here it

seems that the test data (or a large part of them) were part of training the CNN (or classified by

a previous version). I see the risk of introducing a positive bias by effectively preventing (to a

large extent) data that may be classified wrongly by the trained CNN because wrongly classified

particles are excluded in this process of enlarging the training and test dataset. I don't see how

"these images should be considered as ordinary, supervised training images" adequately

discusses this.

I rewrote the whole description of this iterative process, evidencing that misclassified particles were also properly re-labeled and so re-used in the next CNN training. In fact, especially after the first CNN training (based on a very limited data set) a lot of bullets were misclassiifed as columns, rosettes as irregular rosettes, etc.. These images were properly re-labeled to be used for the next CNN training.  In any case, I must say that most of the choices involved in the development of CNNs  (network structure, training parametrs, etc.) are traditionally arbitrary: I describe the method I used for my personal application. The CNN  must  work  with my particular instrument, with its particular illumination, image resolution, etc,  and has not  a widespread application. Moreover, it will be changed in the next future for including more classes of particles as the statistical base increases.

Discussion about importance of sublimation (Re comment on L624/5, 632/633 in previous

review)

L647-649, "This result suggests that sublimation on the DS during the deposition period is less

important than the natural variability of precipitation intensity in determining the number of

particles detected during the acquisition.": I am not sure that you can draw a conclusion like

this based on the evidence you are mentioning. In addition, the following sentence, which should confirm this, is confusing as you seem to refer to lower (colder temperatures) as "temperatures above the median" and warmer temperatures as "in the lower temperature range".

Also L 654-655 suggests that "sublimation does not affect dramatically the number of particles". Again, this cannot be concluded from the results. Contrarily, one must expect that sublimation affects the number of particles (see Fig.9b in Sect 3.2) for smaller particles and at warmer temperatures.

Similarly, the statement in L 691-692 cannot be concluded from the measurements. The resulting size distribution in Fig. 23 may very well be affected by sublimation (and would have been different, with larger and/or more particles, without sublimation). If plates were observed at -50degC or warmer (temperature information is missing in the discussion) then the observed plates may be deposited at the end of the deposition period, shortly before the scan. Particles deposited earlier would have been completely sublimated if less than 200um in size (see Sect 3.2).

Likely, the discussion cannot be improved to support the author's claims. Rather, it should be acknowledged that sublimation cannot be excluded or quantified easily. In addition to particles potentially disappearing due to sublimation (L731-732), particles may also appear smaller than their size during precipitation and when deposited. Statements such as in L738 ("apparently without dramatic losses of small particles for sublimation") cannot be deduced from the presented observations.

I changed the related sentences according with the referee comments.

Other issues:

Resolution of the ICE-CAMERA

Your clarifications refer to the pixel size in the scanned image, but not to the optical resolution.

Can anything be said about the optical resolution? (What are

the smallest details/features that can be seen on the images?)

added the Nyquist criteria

Use of "adverse"

L 307, "The adverse effect is an accelerated natural sublimation of deposited particles." As in the previous review, I have a problem with this sentence and think it is bad language and therefore difficult to follow.

changed

Section numbering

Sect. 4.1.1 is missing?

Renumbered the sections, thank you!

Eventually

L 628, "eventually" is the wrong word.

corrected

Data set or dataset

You use "data set", data-set", and "dataset". In one place also "data store", unclear if you mean

dataset there or something else.

Changed to 'dataset 'everywhere'